human–computer interaction/complexity/mathematical finance

bitcoin, non-parametric statistics, social media, Reddit, text analysis, sentiment

**Author for correspondence:**
Andrew Burnie
e-mail: aburnie@turing.ac.uk

# Social media and bitcoin metrics: which words matter

Andrew Burnie[1,2] and Emine Yilmaz[1,2]

[1]The Alan Turing Institute, London, UK
[2]Department of Computer Science, University College London, London, UK

AB, 0000-0002-8700-3786

We develop a new Data-Driven Phasic Word Identification (DDPWI) methodology to determine which words matter as the bitcoin pricing dynamic changes from one phase to another. With Google search volumes as a baseline, we find that Reddit submissions are both correlated with Google and have a comparable relationship with a variety of bitcoin metrics, using Spearman's rho. Reddit provides complete access to the text of submissions. Rather than associating sentiment with market activity, we describe the DDPWI method for finding specific 'price dynamic' words associated with changes in the bitcoin pricing pattern through 2017 and 2018. We assess the significance of these changes using Wilcoxon Rank-Sum Tests with Bonferroni corrections. These price dynamic words are used to pull out associated words in the submissions thereby providing the context to their use. For example, the price dynamic word 'ban', which became significantly higher in frequency as prices fell, occurred in the context of both government regulation and internet companies banning cryptocurrency adverts. This approach could be used more generally to look at social media and discussion forums at a granular level identifying specific words that impact the metric under investigation rather than overall sentiment.

## 1. Introduction

There has previously been excellent research in analysing the link between Google search volumes and bitcoin metrics, particularly transaction volume and price [1–13]. However, the Google dataset is inherently limited, as it does not allow an in-depth analysis of the words or sentiment underlying a particular search. This article shows that, as an alternative, Reddit matches Google's association with bitcoin metrics, and, in addition, the nature of the Reddit dataset allows for greater functionality when applying data analyses, including identifying words associated with different pricing dynamics.

Bitcoin was originally launched as a challenge to traditional currencies being a 'peer-to-peer version of electronic cash' that

**Figure 1.** The daily bitcoin price in US Dollars from 1 January 2017 to 3 December 2018. The horizontal axis is formatted such that each tick corresponds to the first day of the labelled month. Data sourced from the Charts API of Blockchain Luxembourg S.A. [22].

would enable online payments without the need for intermediates or the oversight of a central bank [14]. However, only 420 retailers in the UK accept bitcoin as a medium of exchange (as of 16 December 2018) [15,16]. Instead, the primary use of bitcoin is now perceived as providing an investment opportunity, and in this respect it has been compared to gold [17,18] and described as a 'crypto-asset' [16]. The weak connect between bitcoin and energy commodity prices, despite the latter having a major influence on the cost of producing bitcoin [19,20], supports the concept that demand for bitcoin is more relevant to price than the cost of supply. Hence, while the value of national currencies is underpinned by a central bank and hard commodities, such as iron and copper, by an intrinsic use (such as in manufacturing), the value of bitcoin is impacted by market sentiment regarding whether bitcoin is perceived as a good investment. Kristoufek [21] links the bitcoin price to transaction data finding that transaction data can explain 88% of price variation. This is consistent with the importance of market sentiment, with changes in opinion leading to bitcoin being bought or sold, resulting in changes to transaction data. The role of sentiment is demonstrated by bitcoin's price dynamics from 1 January to 15 November 2018 (see §1.1 and figure 1), which can be split into three phases of: optimism (prices rising twenty-fold) (Stage 1); pessimism (prices falling to 30% of the peak) (Stage 2); and lastly, a fairly constant oscillation (prices stopped falling) (Stage 3) [22]. Hence, analysing price-sensitive discussions in social media is particularly important for bitcoin.

We examine the specific research problem of determining what was being discussed on social media during the phase of falling prices compared with the phases before (rising prices) and after (stable prices). This requires a new methodology to delineate significant words (§3.3) and the context in which they are being used (§3.4).

## 1.1. Phases in the bitcoin price series

Figure 1 shows the price series and how it splits into three different phases of dynamic

— *Stage 1 (from 1 January to before 16 December 2017)*: Prices rose to 1954.30% of the initial value, from 997.73 to 19498.68 US Dollars.
— *Stage 2 (from 16 December 2017 to before 29 June 2018)*: Prices fell, in a cyclical pattern, to 5908.70 US Dollars (30.30% of the December peak).
— *Stage 3 (from 29 June 2018 to before 15 November 2018)*: Prices traded within a band of 30.30%–42.32% of the highest value in the series (19498.68 US Dollars). The median price, across stage 3, was 6499.06 US Dollars (9.99% above 29 June 2018). Throughout prices remained above the 29 June 2018 value.

After 15 November 2018, the price fell below the 29 June 2018 value and, by the end of the dataset, the price was 3967.52 US Dollars.

## 1.2. Related work

For traditional asset classes, such as equity, a correlation has been shown between Google searches [23] and cumulative weekly stock transaction volume [24] and between searches and stock market moves [25]. Previous bitcoin analyses have also used Google search data [1–13], interpreting internet activity as a proxy for public interest in bitcoin. However, this does not provide any context to the interest, and so is limited in terms of delineating the type of interest. By using Reddit data in conjunction with a new methodology, we are able to determine instead which words are most associated with shifts in the bitcoin price dynamic, and as such we build on existing research by adding a new tool to the existing analytical framework.

Much of the analysis of Google search volumes has been dependent on linear regression [1,2,4–10,12,13,26]. Linear regression assumes that large outliers are unlikely [27], which is inconsistent with the observed recent extreme volatility in bitcoin prices. The median change in prices over 2 years (1 January 2017 to 3 December 2018) was only 0.3247%, but the largest rise was 27.97% on 20 July 2017 and greatest fall was 20.21% on 16 January 2018 [22]. Wavelet analysis has been presented as an alternative [3,28] but this assumed that the different time series compared are normally distributed [29], an assumption not found to hold with bitcoin price series [30,31]. Very few articles [9,10] split the data series into distinct time periods reflecting the phasic pattern of behaviour in bitcoin prices over time, so there is a risk of the results being distorted for any model applied on all data [9].

Knittel & Wash [32] supported using online community text to analyse why users maintain their trust in bitcoin, focussing on Reddit subreddit 'r/bitcoin' because of the higher number and activity of its users compared with the alternatives (see §1.4 for quantitative evidence). Knittel & Wash identified a group of self-described 'Bitcoiners' who refuse to sell bitcoin for currencies backed by a government (e.g. US Dollar) in spite of price fluctuations. Discussions on the subreddit were found to reflect bitcoin-relevant events, with particular concern being over changes to price. This study did not involve statistical analysis, focussed on the limited date range of 3–10 December 2018 and did not examine the link between specific word usage and changes to price.

## 1.3. Contributions of this article

We first show that Reddit submissions and Google search volumes behave in a similar manner. Reddit is studied further because it allows the delineation of the key words, their context and the sentiment of this context, using a VADER analysis [33].

The article builds on earlier work by covering the most recent bitcoin pricing cycle over 2017–2018, with its initial sharp rise, sharp fall and, lastly, a period of relative consistency (figure 1). It, therefore, characterizes how internet and social media behaviour varied with changes in the price of bitcoin during a highly volatile period in bitcoin's price history. We also compare each of the three phases, and are able to determine how the nature of discussions across them shifted focus.

This involves a new Data-Driven Phasic Word Identification (DDPWI) methodology. DDPWI identifies those words whose daily frequencies were statistically significantly higher or lower in a time period of interest compared with the time periods before and after. The periods are selected based on phases observed in a relevant metric. We have observed phases in the bitcoin price (§1.1) and apply DDPWI through comparing the second stage of falling prices with before (when prices rose) and after (when prices stabilized). The resulting 'price dynamic words' are then interpreted, with approaches developed to elucidate the context in which these words were used across the different phases.

Non-parametric approaches are used throughout this article as they require minimal distributional assumptions. DDPWI is based on Wilcoxon Rank-Sum Tests not $t$-tests, and correlation is measured through Spearman's rho not Pearson product-moment correlation or linear regression. Both first assign ranks to the data providing robustness against extreme outliers [34,35] (§1.2). Spearman's rho further tests whether variables move together without requiring the relationship to be linear [27]. This has previously been applied to find statistically significant Spearman's rho correlations between mentions of a company in the Financial Times and associated stock transaction volumes [36].

This article changes the emphasis from testing pre-selected drivers of price (e.g. volume of Google search activity) to identifying price-sensitive words. It focuses on the text within Reddit to determine whether specific words dominate in the communications of individuals at the three different phases outlined above. This analysis moves the debate from ascertaining whether there is a link between the volume of activity and bitcoin price to whether a rise in specific words is indicative of a rising or

falling bitcoin price, or a more stationary phase. As such, we follow the progression of analyses from considering the volume of activity [1,3], to the sentiment of activity [2,7,37], to the actual words used.

## 1.4. Why choose Reddit

In determining which dataset to use, we compared Reddit with a range of public interest sites, word of mouth sites and discussion forums.

### 1.4.1. Public interest sites

Google and Wikipedia are examples of public interest sites, which operate through users inputting search terms to obtain results. They are disadvantaged in that the context of different searches cannot be determined and so the sentiment behind them is unknown. For example, Google results showed increased queries in correlation both with increases in price [1] as well as with the four largest drops in price [2]. Philips & Gorse [28] showed by wavelet analysis that long-term Reddit derived factors were better than Wikipedia searches. Wikipedia entries can be viewed or edited by any internet user with the previous version lost, inhibiting tracking changes in opinions over time [38].

### 1.4.2. Word of mouth sites

The paradigm example of a word of mouth site is Twitter. Twitter text is commonly converted into overall sentiment metrics. Kaminski found a moderate correlation between Twitter sentiments and bitcoin close price and volume [26]. A positive mood on Twitter predicted a bitcoin's price rise 3–4 days later [4], and a positive Twitter sentiment ratio had a positive short-run impact on bitcoin prices [5]. Garcia & Schweitzer [7] found that a higher valence in Tweets preceded increased opinion polarization and exchange volume, which occurred before rises in price. The higher the valence, the greater the degree of pleasure over displeasure in an emotional experience. Polarization measures the extent to which both positive and negative sentiment tweets occurred together. Price drops further led to increased polarization [7].

Abraham et al. [39] argued such studies were flawed in being conducted at a time when prices were continually rising. In their later study, tweet sentiment was found to remain positive overall regardless of the direction of price, sentiment falling below zero only on 1 day in the test group despite 11 of the 19 days (4–24 March 2018) showing price decreases. The use of sentiment classifiers was criticized for describing text as neutral when it may not be in the context of bitcoin. For example, mentioning the current US Dollar bitcoin price is a fact which does not itself carry sentiment. Abraham et al. found that only half the tweets collected on any given day had an objective VADER score, the rest were strictly neutral [39].

The focus of the analysis of Twitter has been on whether the sentiment of tweets can predict the bitcoin price, however, it has not gone further and sought to determine the source of the sentiment, by examining which words dominate. In addition, Twitter identified that 10 million likely fake accounts are created per week, tweeting artificial opinions to skew public perception [40]. Twitter data may thus be unreliable in capturing genuine human perceptions of bitcoin. There is particularly a risk that traders 'pump and dump' [41]: tweet positively about bitcoin to influence perception to encourage a higher price at which to sell. Twitter further lacks Reddit's censorship to ensure that tweets mentioning bitcoin are mainly about bitcoin, which makes it harder to make a meaningful word-based analysis.

### 1.4.3. Discussion forums

Discussion forums, which include Reddit, allow a more sophisticated analysis as there is an element of screening both of content and viewer. There was a statistically significant association between the number of new posts and new members in bitcointalk.org and, in the short-run, subsequent changes in the bitcoin price between 1 November 2009 and 30 September 2013 [9]. Data from the bitcoin forum in bitcointalk.org also showed that positive associations with price were linked to the number of topics posted, positive/very positive comments and positive replies, and that the prediction was highest with a 6-day lag with an accuracy of 79.57% [37].

Reddit is chosen here because of its size. It has 330 million-plus monthly active users and daily there are more than 370 000 posts and 2.8 million comments, with 138 000-plus communities and 80 000-plus

moderators [42]. Data were gathered from the 'Bitcoin' subreddit (https://www.reddit.com/r/bitcoin) where moderators act to ensure that the 'primary topic is Bitcoin' [43]. This distinguishes it from bitcointalk.org where the policy [44] is less stringent regarding discussion of non-bitcoin-related topics. A comparison of the number of online users on 17:40 and 18:08 (25 September 2018; GMT) was, respectively, 1758 and 1643 on bitcointalk.org [45] and 8100 and 8300 on the Reddit subreddit 'Bitcoin' [43]. This comparison was later rerun (19:45, 5 March 2019; GMT) showing 778 online users on bitcointalk.org and 4400 on the Reddit subreddit.

# 2. Data preparation

The dataset extended from 1 January 2017 to 3 December 2018. The data and the code used to prepare and analyse the data are publicly accessible in a Dryad data repository [46]. This is available at: https://doi.org/10.5061/dryad.8n6m564.

## 2.1. Bitcoin metrics

Blockchain Luxembourg S.A. provided the following bitcoin metrics through their 'Charts API' [22]:

— *US Dollar bitcoin price*: This was measured as an average across major exchanges.
— *Bitcoin transaction volume*: The amount of bitcoins transacted on the bitcoin blockchain excluding those returned to the sender as change.
— *US Dollar transaction volume*: The US Dollar value of bitcoins transacted on the blockchain.
— *US Dollar exchange volume*: The US Dollar trading volume of bitcoin on major bitcoin exchanges.
— *Unique, used addresses*: Number of unique addresses used on the blockchain.

The following further metrics were engineered:

— *Bitcoin transaction volume per address*: The transaction volume was divided by the number of unique, used addresses. If transaction volume rises per address, this suggests there is higher activity per user. Metrics were created for US Dollar and bitcoin transaction volume.
— *US Dollar transaction volume per address*.
— *Exchange volume (US Dollar) divided by Transaction Volume (US Dollar)*.
— *Absolute value of daily percentage price change*: Zheng *et al.* [47] found that the use of absolute values was comparable to realized volatility as a measure of market risk.
— *Binary price variable*: This metric equalled one if prices rose on a day and zero otherwise, following Kim *et al.* [11,37]. This helped compensate for extreme outliers that could skew a correlation with percentage price change.

## 2.2. Google search

Google search volumes on the topic of bitcoin (labelled 'Bitcoin - Cash') were available for download in CSV format from Google Trends [23]. Specifying the range of dates to be 1 January 2017 to 3 December 2018 resulted in data being provided on a weekly basis. Therefore, in order to be able to extract Google search volumes on a daily basis, we split the range of dates into blocks. The data in each block were reported relative to the maximum value in that block (set to equal 100) [23], and so each block was of a different scale. This prevented comparison of search volumes across blocks. The rescaling technique followed Kristoufek [3].

Data were downloaded in sequential 8-month blocks where the last four months of a block overlapped with the first four of the next chronologically. The exception was the last block which was from 1 May 2018 to 3 December 2018. The values in the last block were rescaled such that all values within were relative to the final data-point (3 December 2018), which was set to equal 100.

The values in all the other blocks were rescaled to be relative to this final data-point. In the penultimate block (1 January 2018 to 1 September 2018), the last four months (1 May 2018 to 1 September 2018) overlapped with the first four of the last block (1 May 2018 to 3 December 2018). The multiplier values required to convert each Google search volume in the earlier block to the differently scaled value on the same date in the last block were found. These multipliers varied slightly across dates and so the harmonic mean was calculated. The harmonic mean was selected because it resulted in the lowest

average error compared with arithmetic mean, median, mode and geometric mean. This harmonic mean multiplier was multiplied to the values in the earlier block that lacked corresponding dates (1 January 2018 to 30 April 2018) in the latter block; the rescaled values were then concatenated to the values in the latter block, extending the range of dates covered (now 1 January 2018 to 3 December 2018). The process was repeated for the second to last block (1 September 2017 to 1 May 2018), third to last (1 May 2017 to 1 January 2018) and the earliest block (1 January 2017 to 1 September 2017).

The result was daily Google search volumes available from 1 January 2017 to 3 December 2018.

## 2.3. Reddit submissions

We used the Pushshift API [48] to extract the text for each submission to the 'Bitcoin' forum. Submissions data were selected over comments because the latter were prone to deviate onto arguments on bitcoin-irrelevant topics, such as religion, non-specific insults and different date formats (https://www.reddit.com/r/Bitcoin/comments/9svjcp/10_years_ago_today_2008_oct_31/). The number of submissions per day were the number that remained after text processing (§2.4).

## 2.4. Engineering word frequency data from Reddit submissions text

The submissions were filtered and the text processed and tokenized to produce word lists.

### 2.4.1. Submission filtering

The following submissions were filtered out: those authored by 'rBitcoinMod', as these consisted primarily of automated text stating forum guidelines for the 'Daily Discussion' and 'Mentor Monday'; those authored by 'crypto_bot', as these consisted mainly of automated, daily data updates on the bitcoin network; submissions with identical text to another submission; blank submissions; and submissions that had been entirely removed, thus whose text consisted of only '[deleted]' or '[removed]' [49].

### 2.4.2. Text pre-processing

(i) All text was put into the lower case.
(ii) The accepted currency codes [50] 'btc' and 'xbt' were converted into the synonymous 'bitcoin'.
(iii) The following were removed respectively: strings of 50 or more consecutive word characters (as this is too long to represent a word); URLs; HTML tags (e.g. '&'); the new line character ('\n'); Twitter (e.g. '@john') and Reddit handles (e.g. '/u/john' and '/r/john'); references to deleted text ('[removed]' and '[deleted]'); and non-ASCII text (e.g. Cyrillic alphabet or emoticons).
(iv) The US Dollar was referred to in 11.30% submissions as: '$', 'usd', 'dollar(s)' and 'us dollar(s)'. These were treated as synonymous and were all replaced by 'dollar_marker_symbol'.
(v) Punctuation and apostrophes were removed unless these were inside words to indicate abbreviations (e.g. 'o'clock').
(vi) 'tx' was used to abbreviate [51] and thus was replaced by the word 'transaction'. Both 'ln' and 'lightning network' were replaced with 'ln'. The terms 'telephone number' and 'phone number' were replaced with 'phone_number'.

### 2.4.3. Daily word frequencies

Text was converted into word lists using Python package NLTK v.3.3 and its associated download 'punkt'. NLTK removed 'stopwords' which were high-frequency words unrelated to a particular topic. The term 'n't' was included as an abbreviation for the stopword 'not'.

Words with the same meaning but different grammatical case were combined. Each word was lemmatized using NLTK's 'WordNetLemmatizer'. The context of a word was determined by looking it up in a dictionary and mapping different cases of the same word to a base form. This failed for unusual words (e.g. 'bitcoins' and 'bitcoin', and 'ICO' and 'ICOs') that were not in the dictionary. Hence, stemming was subsequently applied using 'SnowballStemmer'. This applied to all words a set of rules that ignore the context of the word, and so was extendable to rare words. The 'snowball' stemmer was chosen as it is the least likely to treat words of the same concept differently or words of a different concept the same [52]. For cryptocurrency mining, two abbreviations 'miner' and 'mine' were merged into 'mine'.

To prevent skewing by a few longer submissions, each word was counted once if present in a given submission. Words in 100 or less submissions were removed. There were 326 945 submissions with 131 656 words of which 3900 were found in more than 100 submissions. A 'day' was specified to be from 00:00 on a given day to before 00:00 on the next date (GMT).

Daily counts of the 3900 words were normalized by dividing the count by the daily total number of submissions to ensure that word frequency measured the proportion of submissions containing a term.

# 3. Methodology

## 3.1. Correlation analyses of Google, Reddit and Bitcoin metrics

Calculating the correlation between a pair of values involves measuring the extent to which, as one variable rises, the other also rises (positive correlation) or falls (negative correlation). Spearman's rho measures such monotonic relationships, without requiring that the association be linear and without assuming each variable follows a normal distribution, in contrast to Pearson Product-Moment Correlation [34]. Previous studies have found bitcoin prices to follow a non-normal distribution [31,53].

Measuring Spearman's rho involves separately ranking two sets of values ($x$ and $y$) in terms of magnitude ($R_x$ and $R_y$). Equation (3.1) is then applied (where $\bar{a}$ is the arithmetic mean value for variable $a$) [54]. This results in a value varying from $-1$ (the variables are perfect opposites) and 1 (the values follow each other perfectly). Statistical significance was evaluated through two-sided tests.

$$\frac{\sum (R_{xi} - \bar{R}_x)(R_{yi} - \bar{R}_y)}{\sqrt{\sum (R_{xi} - \bar{R}_x)^2 \sum (R_{yi} - \bar{R}_y)^2}}. \tag{3.1}$$

Correlation analyses were applied to evaluate the similarities between Reddit and Google. This involved calculating correlations between Reddit submissions (§2.3) and Google search volumes (§2.2), and between these and the ten bitcoin metrics (§2.1). In addition, we assessed how Reddit submissions and Google searches were associated with the values of the bitcoin metrics recorded 1 and 2 days after the Reddit and Google volume metrics.

If the values compared follow random walks, this violates Spearman's rho's assumption that the different pairs of values are independent [55]. The original values were thus converted into the daily percentage change in value to provide robustness [27,56].

Augmented Dickey-Fuller (ADF) Tests assessed the risk of a random walk. These were applied at a 5% significance level, with an intercept and lags determined by Akaike information criteria [27]. The ADF Test assumes no structural breaks [57], and so these tests were applied within each identified stage of pricing behaviour figure 1. The ADF Tests supported there being sufficient evidence to reject a null hypothesis of a random walk for Google search volumes, Reddit submission volumes and all the bitcoin metrics, with all $p$-values being below 5%. These tests were run on the daily percentage change in value with the exception of the 'Binary Price Variable', which was already based on the change in price (see §2.1).

Statistical robustness was further enhanced through applying a Bonferroni correction which divides the $p$-value cut-off by the number of tests run [58]. For the 10 bitcoin metrics, the 1% $p$-value cut-off was divided by 10, resulting in a threshold $p$-value of 0.001. Correlation analyses were run using both the entire dataset and within each phase.

## 3.2. Identifying words by absolute frequency

The words that were in at least 5% submissions in any one of the identified stages of the bitcoin price series were first identified. This was to determine the extent certain words dominated discussions across time.

## 3.3. Identifying words by relative frequency

### 3.3.1. Comparing word frequencies across stages

A methodology was required to statistically evaluate for which words the daily frequencies were typically higher or lower in one stage of the price series compared with the previous stage (§1.1).

Extreme outliers were present, even for a popular word such as 'bitcoin', which never fell below 35% submissions on a given day. Across three days, the popularity of 'bitcoin' fell from 46.75% submissions (19 July 2017) to 38.58% (20 July 2017) to recover to 48.71% the next day (21 July 2017). This precluded using the $t$-test in comparing daily word frequencies across price phases, as this is sensitive to extreme outliers [35].

Instead, the non-parametric equivalent, the two-sided Wilcoxon Rank-Sum Test, was used to delineate which words had daily frequencies that had changed significantly across different phases in the price series. An additional Bonferroni correction, such that the $p$-value cut-off (1%) was divided by the number of tests (3900), ensured that the identification of significant words was robust [58].

### 3.3.2. Applying DDPWI to identify price dynamic words

The DDPWI approach identifies those words where the change in frequency from phase 1 to 2 (rising prices shifting to falling) and from phase 2 to 3 (falling prices ceasing to fall further) are opposite and both statistically significant. The words that changed statistically significantly were identified using the two-sided Wilcoxon Rank-Sum Test and restricted to those with above 1% frequency in phase 2. We define the words resulting from applying DDPWI as the 'price dynamic' words.

## 3.4. Context of price dynamic words

### 3.4.1. Identifying the context

An iterative procedure for generating the theme of a typical sentence that contained one of the price dynamic words was developed:

  (i) Let $W$ represent a chain of words. Initially, $W = [w_1]$, where $w_1$ was the specific word of interest.
  (ii) Extract only submissions that contain all words in $W$.
  (iii) Find the most frequent word in these submissions and append to $W$.
  (iv) Repeat (ii)–(iii) until $W$ is of length 5, excluding the word of interest, or there exists at least two words of the same highest frequency in step (ii).
  (v) The result was a chain of related words, $W = [w_1, w_2, \ldots]$.

Every iteration reduced the number of submissions considered. Generic words (e.g. 'bitcoin' and 'would') that provided little thematic content and synonyms were censored.

### 3.4.2. Sentiment of the context

Sentiment was measured, using the VADER [33] algorithm, for submissions that contained the price dynamic words, using 'bitcoin' as a control. VADER was designed for social media text and so is able to handle both emoticons and slang [33,37]. Text processing was thus minimized to converting 'tx' (a bitcoin-specific abbreviation [51]) into 'transaction' and removing tokens that should not have a sentiment (e.g. URLs, HTML tags and '[deleted]'). Individual submissions with a compound sentiment score of less than − 0.2 were labelled as 'negative' and those with a score of at least 0.2 as 'positive' [37]. The number of positive sentiment submissions was divided by the number of positive and negative sentiment submissions to derive the positive sentiment metric. The negative sentiment metric was similarly normalized. The sentiment metrics were calculated for submissions across the past 90 days to prevent noise in the metric from obscuring the identification of underlying trends in the sentiment over time.

# 4. Results

## 4.1. Reddit submissions descriptive statistics

Table 1 presents descriptive statistics on Reddit submissions, showing a decline in Reddit activity as prices stabilized. On average, over 500 submissions were posted per day when prices were most volatile in stages 1 and 2; this fell 46% with stage 3.

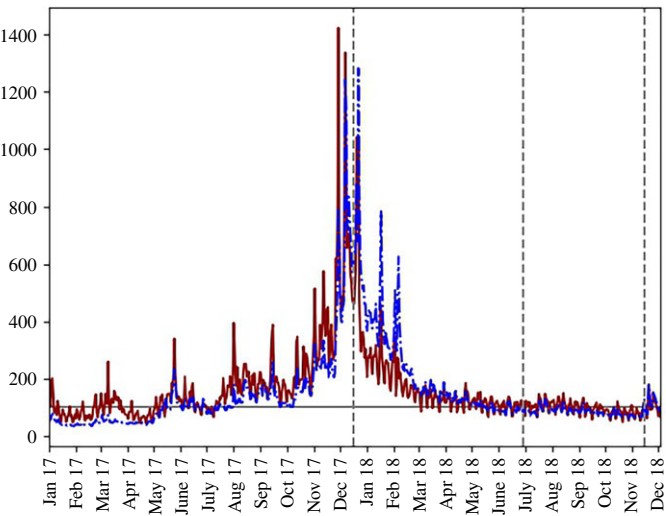

**Figure 2.** Daily Reddit submission volume (red, solid line) and Google search volume for the bitcoin topic (blue, dot-dash line) from 1 January 2017 to 3 December 2018. All values are relative to the last value (3 December 2018) which was scaled to equal 100. The grey horizontal line indicates the base value of 100; the vertical dashed lines indicate the boundaries of the identified stages of pricing dynamic (see §1.1). The horizontal axis is formatted such that each tick corresponds to the first day of the labelled month. Google data sourced from Google Trends [23] and Reddit data from Pushshift API [48]. See §2 for details on data preparation.

**Table 1.** Descriptive statistics for Reddit submissions for across the dataset (1 January 2017 to 3 December 2018) and within stages 1, 2 and 3 (see §1.1).

| stage | days | submissions | submissions per day |
|---|---|---|---|
| all data | 702 | 326 945 | 465.73 |
| 1 | 349 | 181 327 | 519.56 |
| 2 | 195 | 101 110 | 518.51 |
| 3 | 139 | 38 706 | 278.46 |

## 4.2. Comparing Reddit submissions with Google searches

Figure 2 shows, graphically, how Google search volumes and Reddit submissions followed similar patterns across the dataset. Both rose to a peak in November–December 2017 and then sharply fell back before slowing and then finally reaching a relatively constant level by June 2018. During the time of highest activity both series experienced spikes where values doubled to then quickly revert.

The close association between Google searches and Reddit was supported by a positive Spearman's rho correlation (0.6879) between the percentage change in Google searches and the percentage change in Reddit submissions. This correlation was statistically significant, with the associated two-sided test having a low $p$-value ($2.23 \times 10^{-99}$). This result was robust to splitting the dataset up into stages: stage 1 correlation was 0.6701 ($p$-value of $1.07 \times 10^{-46}$); stage 2 correlation was 0.6741 ($p$-value of $3.43 \times 10^{-27}$); and stage 3 correlation was 0.7131 ($p$-value of $6.94 \times 10^{-23}$).

## 4.3. Web metrics correlated with Bitcoin metrics

The pattern of results was similar for both Google and Reddit against the 10 bitcoin metrics (§2.1), with a significant positive correlation between internet activity and the bitcoin metrics on the same day table 2.

Positive correlates occurred on the same day for both Google and Reddit with the bitcoin transaction volume, the US Dollar transaction volume, the US Dollar exchange volume, the unique and used addresses, bitcoin and US Dollar transaction volume per such address, the exchange divided by transaction volume and the absolute variation in bitcoin price. There was no correlate with the bitcoin US Dollar price.

When subdivided into the three stages the same day positive correlations were maintained in stages 1 and 2 for Google searches and Reddit submissions with the exceptions of the absolute variation in price

**Table 2.** Spearman's rho correlation between percentage change in daily internet activity metric (Google searches for the bitcoin topic or Reddit submissions) and bitcoin-related metrics ($V$) with associated two-sided significance test $p$-values across all data (1 January 2017–3 December 2018), and within stages 1, 2 and 3 (see §1.1). The metrics considered are US Dollar price ($V = 1$); bitcoin transaction volume ($V = 2$); US Dollar transaction volume ($V = 3$); US Dollar exchange volume ($V = 4$); Unique and Used Addresses ($V = 5$); bitcoin transaction volume per such addresses ($V = 6$); US Dollar transaction volume per such addresses ($V = 7$); exchange divided by transaction volume ($V = 8$); binary variable equal to one if price change >0 else zero ($V = 9$); and absolute variation in price ($V = 10$). These metrics were expressed in terms of daily percentage change except for $V = 9$ (see §4.3). *Indicates significant at the Bonferroni-corrected 1% significance level.

| $V$ | all | $p$-value | stage 1 | $p$-value | stage 2 | $p$-value | stage 3 | $p$-value |
|---|---|---|---|---|---|---|---|---|
| Google | | | | | | | | |
| 1 | −0.0418 | 0.2690 | 0.0838 | 0.1185 | −0.2247 | 0.0016 | −0.0307 | 0.7199 |
| 2 | 0.5363* | <0.0001 | 0.4442* | <0.0001 | 0.5639* | <0.0001 | 0.6677* | <0.0001 |
| 3 | 0.5220* | <0.0001 | 0.4415* | <0.0001 | 0.5244* | <0.0001 | 0.6690* | <0.0001 |
| 4 | 0.6255* | <0.0001 | 0.5862* | <0.0001 | 0.6517* | <0.0001 | 0.6624* | <0.0001 |
| 5 | 0.4361* | <0.0001 | 0.2955* | <0.0001 | 0.5692* | <0.0001 | 0.5816* | <0.0001 |
| 6 | 0.4380* | <0.0001 | 0.3513* | <0.0001 | 0.4006* | <0.0001 | 0.6139* | <0.0001 |
| 7 | 0.4213* | <0.0001 | 0.3541* | <0.0001 | 0.3419* | <0.0001 | 0.6166* | <0.0001 |
| 8 | 0.2805* | <0.0001 | 0.4152* | <0.0001 | 0.3071* | <0.0001 | −0.1057 | 0.2157 |
| 9 | −0.0420 | 0.2673 | 0.0509 | 0.3442 | −0.2059 | 0.0039 | −0.0061 | 0.9432 |
| 10 | 0.2516* | <0.0001 | 0.1723 | 0.0013 | 0.2356* | 0.0009 | 0.4129* | <0.0001 |
| Reddit | | | | | | | | |
| 1 | −0.0262 | 0.4884 | 0.1050 | 0.0504 | −0.2019 | 0.0046 | −0.0525 | 0.5391 |
| 2 | 0.6058* | <0.0001 | 0.5574* | <0.0001 | 0.6168* | <0.0001 | 0.6943* | <0.0001 |
| 3 | 0.5951* | <0.0001 | 0.5611* | <0.0001 | 0.5759* | <0.0001 | 0.6869* | <0.0001 |
| 4 | 0.6036* | <0.0001 | 0.6400* | <0.0001 | 0.5623* | <0.0001 | 0.5269* | <0.0001 |
| 5 | 0.4724* | <0.0001 | 0.2810* | <0.0001 | 0.6205* | <0.0001 | 0.7129* | <0.0001 |
| 6 | 0.5174* | <0.0001 | 0.5210* | <0.0001 | 0.4318* | <0.0001 | 0.5874* | <0.0001 |
| 7 | 0.5037* | <0.0001 | 0.5254* | <0.0001 | 0.3716* | <0.0001 | 0.5852* | <0.0001 |
| 8 | 0.1874* | <0.0001 | 0.3996* | <0.0001 | 0.1339 | 0.0620 | −0.2735 | 0.0011 |
| 9 | −0.0376 | 0.3207 | 0.0349 | 0.5163 | −0.1669 | 0.0197 | −0.0183 | 0.8308 |
| 10 | 0.2528* | <0.0001 | 0.2775* | <0.0001 | 0.2378* | 0.0008 | 0.1578 | 0.0636 |

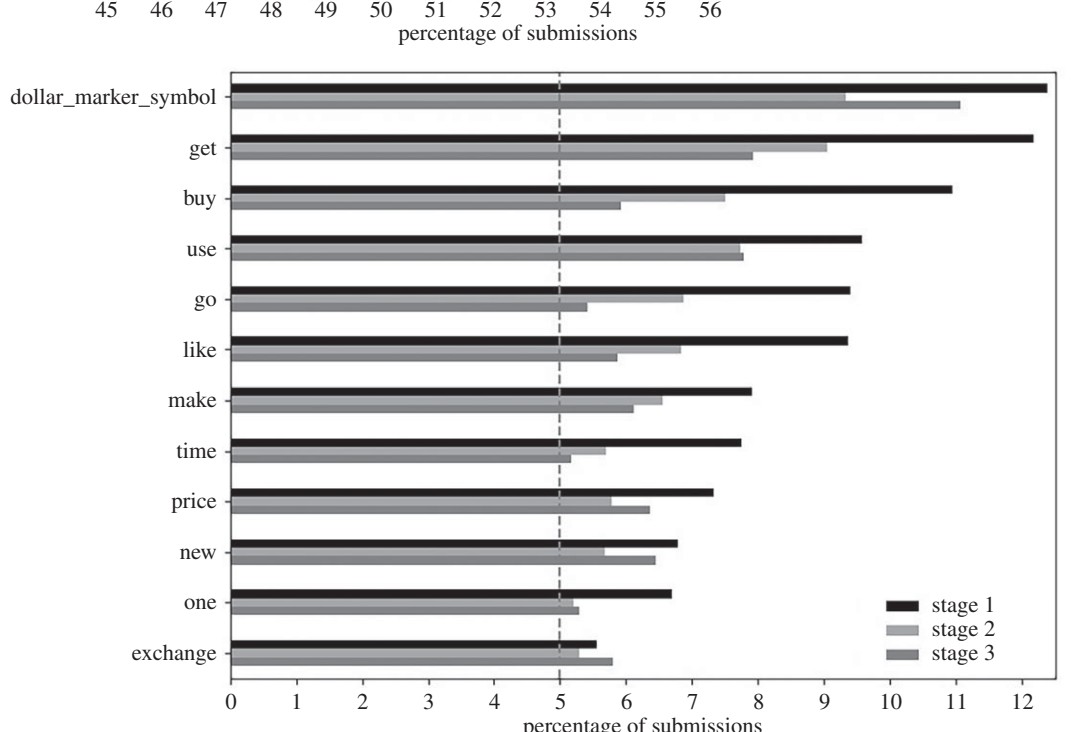

**Figure 3.** Words in at least 5% submissions in all stages, and the percentage of submissions they were in for each stage. Bitcoin is graphed separately because it was more than twice as frequent as the next word. The dashed, vertical grey line represents the 5% cut-off. The top bar represents the percentage of submissions containing the term in stage 1; the middle bar is the percentage in stage 2; and the bottom bar is the percentage in stage 3. Each 'word' is a lemmatized and then stemmed version of the original word. For example, 'exchang' represents exchange, exchanges, exchanged and exchanging, and 'use' represents 'use', 'uses', 'used' and 'using'. The term 'dollar_marker_symbol' represents different synonyms for the US Dollar (see §2.4).

(Google) and exchange divided by transaction volume (Reddit). In stage 3, the findings were maintained for all the metrics on the same day except for the exchange divided by transaction volume (Google and Reddit) and the absolute variation in price (Reddit).

Correlating internet activity with the bitcoin metric values one and two days after resulted in findings whose statistical significance was not robust to dividing the data into stages.

## 4.4. Most frequent words by absolute frequency

Figure 3 lists those words in at least 5% (one in twenty) submissions in all stages. The term 'bitcoin' was the commonest, in about half of submissions. The other terms conveyed the persistent popularity of discussion around the bitcoin price ('dollar_marker_symbol' and 'price'), acquiring bitcoin('get', 'buy', 'make'), opinions (like), innovation (new) and exchanges (exchang). All these terms had a statistically significant fall from stages 1 to 2 except 'exchang' ($p$-value $1.95 \times 10^{-1}$). The 'dollar_marker_symbol' term rose significantly in popularity from stages 2 to 3 ($p$-value $4.94 \times 10^{-12}$).

Figure 4 lists those words that were in at least 5% submissions in an incomplete number of stages. Twenty fell significantly from stage 1 to 2 with 14 falling to below the 5% threshold. 'Blockchain' became popular in stage 3 (46.12% rise on stage 2, $p$-value of $3.39 \times 10^{-10}$) and so did'market' (48.14% rise on stage 2, $p$-value of $1.72 \times 10^{-17}$). Cryptocurrency discussions more than doubled in frequency from phase 1 to 2, an upward trend that continued to phase 3. The term 'coinbas[e]' (frequency of 5.83% in stage 1) referred to the cryptocurrency exchange Coinbase (https://www.coinbase.com/).

## 4.5. Comparing word frequencies across stages and identifying price dynamic words

Eleven words demonstrated a statistically significant change in frequency when moving from both phase 1 to 2 and phase 2 to 3.

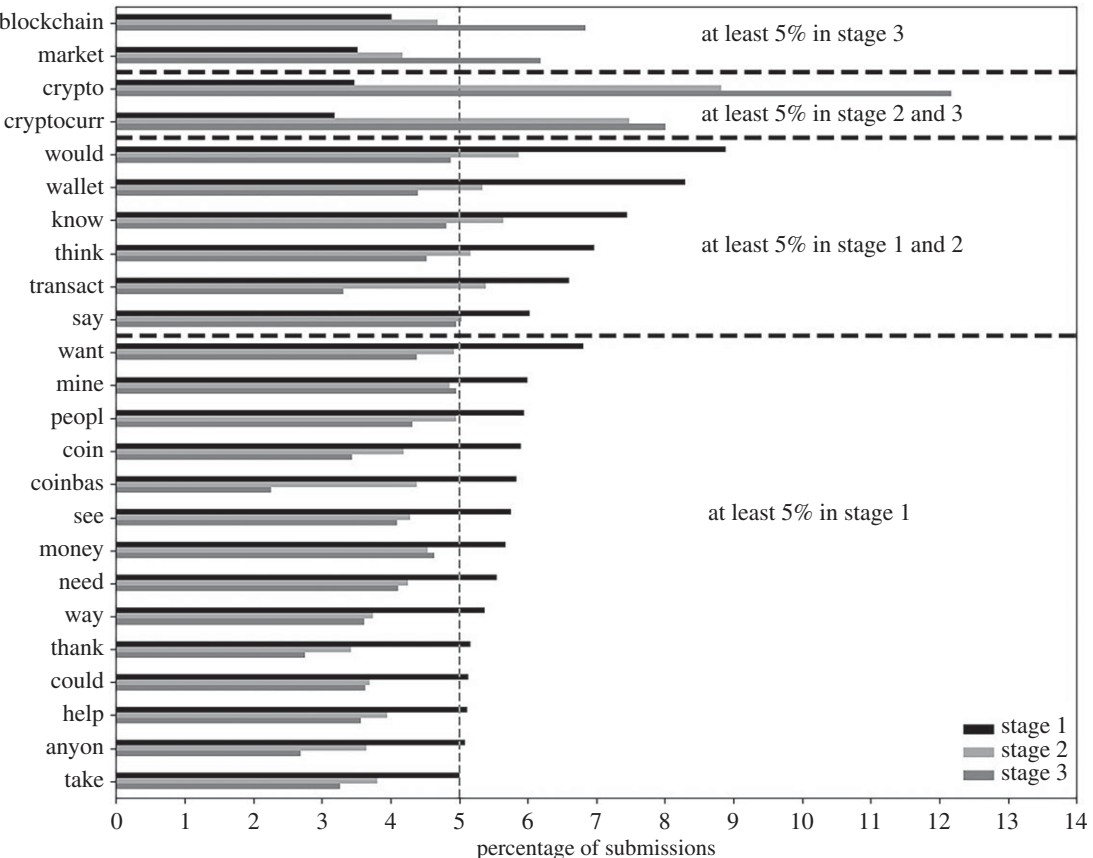

**Figure 4.** Words in at least 5% submissions in at least one stage but not all, and the percentage of submissions they were in for each stage. These words consisted of four groups demarcated by the black, horizontal lines: those words in at least 5% submissions in stage 1 alone (bottom words); in stages 1 and 2 (penultimate from bottom); in stages 2 and 3 (penultimate from top); and in stage 3 alone (top). The dashed, vertical grey line represents the 5% cut-off. The top bar represents the percentage of submissions containing the term in stage 1; the middle bar is the percentage in stage 2; and the bottom bar is the percentage in stage 3. Each 'word' is a lemmatized and then stemmed version of the original word. For example, 'exchang' represents exchange, exchanges, exchanged and exchanging, and 'use' represents 'use', 'uses', 'used' and 'using'.

Six words rose across both phasic shifts: 'investor', 'market', 'million', 'crypto', 'launch' and 'platform'. Two words fell across both phasic shifts: 'segwit' and 'fee'.

Three words fulfilled the definition of a price dynamic word (§3.3) in that the change in frequency was opposite and statistically significant from phase 1 to 2 and from phase 2 to 3: 'tax' and 'ban' rose from stage 1 to 2 and fell from stage 2 to 3; while 'dollar_marker_symbol' fell from stage 1 to 2 and rose from stage 2 to 3.

## 4.6. Context of price dynamic words

### 4.6.1. Identifying the context

The word 'ban' occurred most with 'china' and 'exchang[es]' in stage 1 table 3 but these associated words did not continue into stages 2 or 3. In stage 2, bans were mentioned in the context of 'central' 'bank' 'cryptocurr[ency]' regulation. A subanalysis of the ten most frequent words associated with 'ban' demonstrated 'trade' (11.98%) and 'ad' (11.20%) were specific to stage 2, and 'googl[e]' (12.75%) was specific to stage 3. When 'ban' and 'trade' were run together (stage 2), 'korea' was the most frequent word (42.21%). When 'ban' and 'ad' were run together, the chain of associations were: 'facebook' (42.36%), 'googl[e]' (22.95%) and then 'twitter' (78.57%). In stage 3, 'ban[s]' by the 'india[n]' 'reserv[e]' 'bank' became a topic. When 'ban' was paired with 'googl', 'ad' had the highest frequency (46.15%). In submissions with these three words, 'end' (50.00%) was the most frequent.

US Dollars were discussed the most with the word 'buy' in stages 1 and 2. This pair was mentioned more with 'price' in stage 1 and 'sell' in phase 2. Phase 3 was distinct—'price' (20.63%) was mentioned

**Table 3.** Chain of most frequent words associated with price dynamic words: 'tax', 'ban' and 'dollar_marker_symbol'. At each step, submissions were reduced to those containing all previous words in the chain and then the most frequent word in these submissions was found and expressed as a percentage of submissions. For example, starting with stage 1 submissions containing the word 'tax', the most frequent word was 'pay' (34.17% of those submissions). The word that was most frequent in submissions with the words 'tax' and 'pay' was 'buy', in 35.12% of these submissions. In submissions that contained 'tax', 'pay' and 'buy', 48.73% contained the word 'sell'.

| stage | chain |
|---|---|
| | 'ban' |
| 1 | 'china' (38.50%) - 'exchang[e]' (30.20%) - 'price' (31.69%) - 'peopl[e]'/'would'/'time'/'trade' (55.17%) |
| 2 | 'cryptocurr[ency]' (29.63%) - 'bank' (23.10%) - 'central' (37.50%) - 'govern' (36.36%) - 'time'/'technolog[y]' (75.00%) |
| | **Starting with 'ban' and 'trade', censoring 'cryptocurr[ency]' and 'crypto'** |
| | 'korea' (42.21%) - 'south' (83.08%) - 'say'/'plan' (20.37%) |
| | **Starting with 'ban' and 'ad', censoring 'cryptocurr[ency]' and 'crypto'** |
| 3 | 'facebook' (42.36%) - 'googl[e]' (22.95%) - 'twitter' (78.57%) - 'plan' (36.36%) |
| | 'cryptocurr[ency]' (27.12%) - 'india' (31.33%) - 'bank' (34.62%) - 'reserv[e]'/'court' (44.44%) |
| | **Starting with 'ban' and 'googl[e]', censoring 'cryptocurr[ency]' and 'crypto'** |
| | 'ad' (46.15%) - 'end' (50.00%) - 'month' (44.44%) - 'next'/'news' (75.00%) |
| | 'dollar_marker_symbol' |
| 1 | 'buy' (24.05%) - 'price' (27.75%) - 'time' (39.43%) - 'one' (47.38%) - 'peopl[e]' (63.57%) |
| 2 | 'buy' (18.66%) - 'sell' (30.66%) - 'price' (47.22%) - 'peopl[e]'/'time' (49.80%) |
| 3 | 'price' (20.63%) - 'market' (26.02%) - 'time' (50.00%) - 'exchang[e]'/'trade' (54.78%) |
| | 'tax' |
| 1 | 'pay' (34.17%) - 'buy' (35.12%) - 'sell' (48.73%) - 'gain' (55.21%) - 'capit[al]' (75.47%) |
| 2 | 'pay' (28.95%) - 'gain' (36.13%) - 'capit[al]' (61.29%) - 'year'/'sell' (45.26%) |
| 3 | 'pay' (25.57%) - 'one'/'would' (40.30%) |

more than 'buy' (13.70%) and dollars and price were mentioned most frequently with the word 'market' (26.02%).

The word 'tax' occurred most frequently in association with the word 'pay' throughout all three stages. Associated with 'pay[ing]' 'tax', was 'capit[al]' 'gain[s]' (stages 1 and 2) and 'buy[ing]' (stage 1) and/or 'sell[ing]' (stages 1 and 2) bitcoin.

### 4.6.2. Sentiment of the context

Examining bitcoin mentions figure 5 demonstrated that positive sentiments were more than twice as frequent than negative across all three stages. Sentiment initially became more negative during the phase of falling prices, but from March 2018 this trend reversed. Overall bitcoin mentions fell during phase 2, from over 58% to below half of submissions, but from April 2018 onwards this reversed.

Figure 6 shows that twice as many 'ban' submissions were negative than positive, and there was a drift towards more negative sentiment over time. There were periods of particularly high interest where frequency was above 1.6%: the 90 days up to October–November 2017 (phase 1), and in April 2018 and June 2018 (both phase 2).

Similar to bitcoin, the frequency of US Dollar mentions fell at the start of phase 2 with this trend reversing from April 2018 figure 7. Sentiment was twice as positive than negative across the three

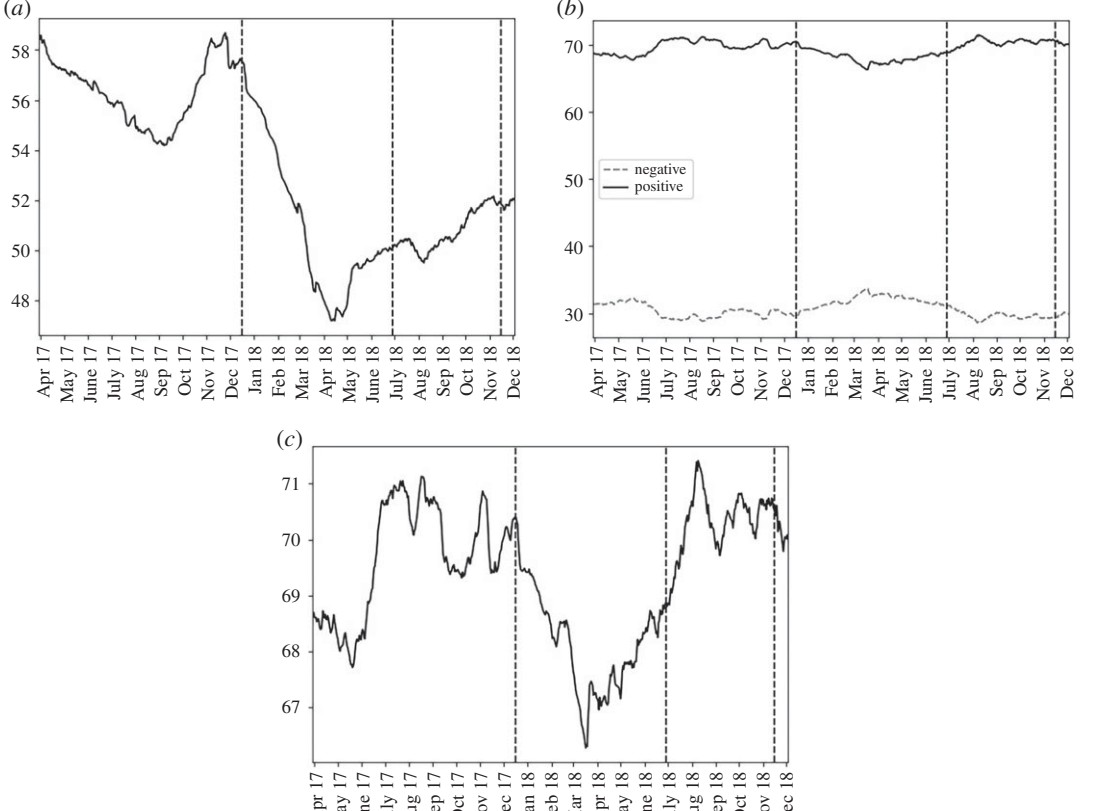

**Figure 5.** For 'bitcoin', over the past 90 days, from the top left panel clockwise: (*a*) the percentage of submissions containing the term; (*b*) the percentage of negative and positive submissions that were of positive sentiment (solid, black line) or negative sentiment (dashed, grey line); (*c*) the percentage that were of positive sentiment.

phases. Sentiment became more negative during phase 2 and, unlike with bitcoin, this trend reversed only with the shift from phase 2 to 3.

Interest in 'tax' figure 8 began to rise just before phase 2, more than doubling in frequency from less than 0.8% (90 days to November 2017) to fluctuating around 1.6% submissions (March–May 2018, phase 2). Frequency subsequently fell to about 0.6% by August 2018 (phase 3). There were more than 2.5 times as many positive than negative submissions mentioning 'tax' across the dataset.

# 5. Discussion

This article establishes that Reddit submissions are similar to Google searches in capturing internet activity but are an improvement in providing greater textual content for more in-depth analyses. Reddit submissions' strong correlation with Google searches suggests that either redditors and Google searchers react similarly to external events or the same people are interacting with one website and then the other. Reddit submissions had a comparable relationship with a variety of bitcoin metrics. In addition, greater internet activity was associated with both greater numbers of users (measured by unique bitcoin addresses used per day) as well as more buying and selling from existing users (transaction volume divided by addresses).

Word frequency analysis shows the evolving nature of Reddit discussions over the three phases. This is first supported through examining the most frequent words (figures 3 and 4). During stage 1, discussions were more orientated towards people considering entering the bitcoin network, thus the particularly high popularity of 'get', 'buy', 'want', 'wallet' and 'mine', and the exchange Coinbase was more frequently considered than in subsequent periods. These words became less popular during stages 2 and 3. During stages 2 and 3, the frequency of submissions discussing crypto and cryptocurrencies more than doubled that in stage 1; likewise, there was an uplift in discussion of blockchain. This would be consistent with interest broadening from bitcoin to other forms of cryptocurrencies and their associated blockchain technology.

Certain words changed statistically significantly in frequency between one phase and the next (§4.5). For example, there was a decline in the debate concerning the 'segwit' bitcoin fork, whereas there was rising popularity in trading ('investor', 'market') and cryptocurrency innovation ('crypto', 'launch').

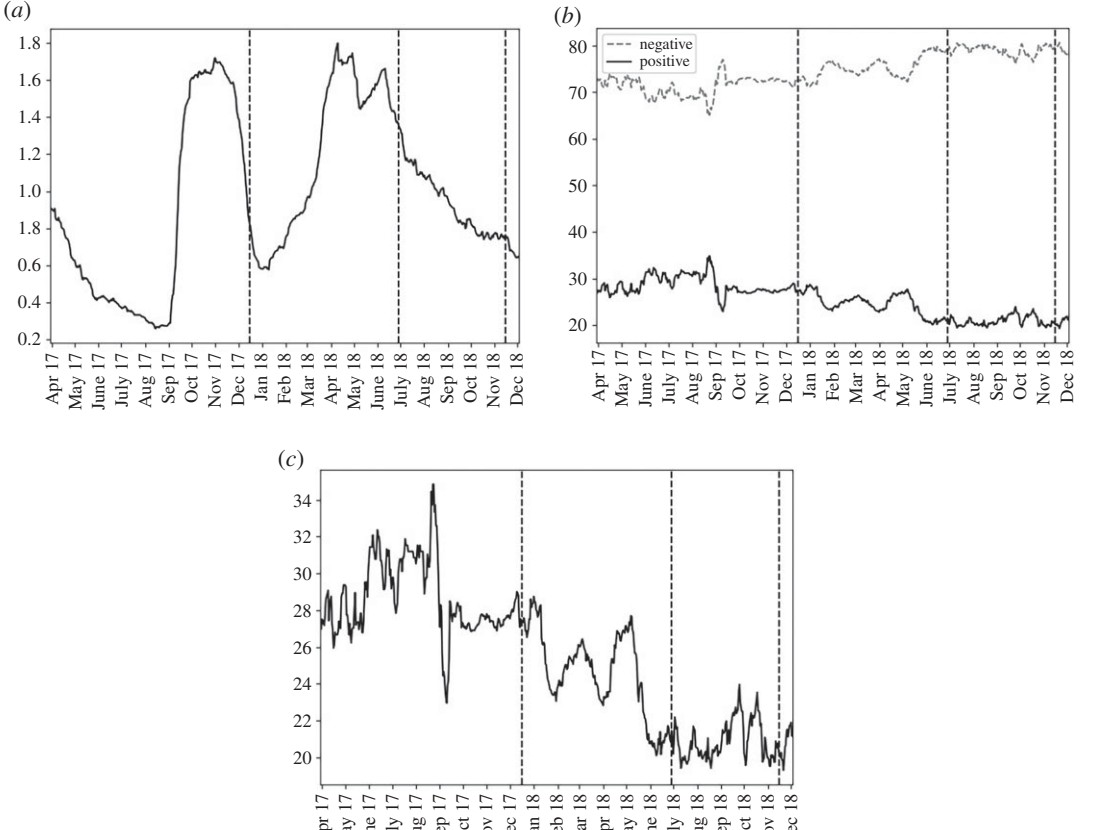

**Figure 6.** For 'ban', over the past 90 days, from the top left panel clockwise: (*a*) the percentage of submissions containing the term; (*b*) the percentage of negative and positive submissions that were of positive sentiment (solid, black line) or negative sentiment (dashed, grey line); (*c*) the percentage that were of positive sentiment.

Applying DDPWI identified three 'price dynamic' words whose frequencies were statistically significantly higher or lower during the phase of falling prices (phase 2) than before (phase 1) and after (phase 3): ban, tax and US Dollars.

The decision of whether or not to implement a ban through government regulation or corporate policy is an exogenous event not directly dependent on changes in the bitcoin price, thus the higher frequency of 'ban' as prices fell was not likely to be caused by the falling prices. A more plausible explanation is that it was announcements, implementations or rumours of bans, which resulted in related discussions on social media, that influenced the reductions in price.

The word 'ban' occurred in a shifting context (table 3) of consistently negative sentiment. This context changed from regulation in China (phase 1) to South Korea (phase 2) to India (phase 3), while discussions about internet company bans on adverts became evident only in phases 2 and 3. Discussions of 'bans' became particularly frequent from September to November 2017 (just before phase 2) and rose in frequency from January 2018 to a peak in April 2018 (in phase 2) figure 6. Higher concern over bans coincided with speculation of or actual bans being implemented. For example, the phase 1 activity occurred with China announcing a ban on exchanges in September 2017 with the last exchange closing in November [59]. In previous studies covering earlier time periods, the effect of 'China' on the bitcoin price has been suggested [3,8] and, using topic modelling, the concept 'China' was predictive towards the bitcoin price [11].

During phase 2, there was speculation as to the extent to which cryptocurrency activities would be banned in South Korea [60]. Facebook banned cryptocurrency adverts from January to June 2018 [61], followed by announcements of bans by Twitter [62] and Google [63] in March 2018. The chain of word frequencies in table 3, stage 3, could be explained by a court decision, in India, to uphold the cryptocurrency ban, made in July 2018 [64], and Google's ban on cryptocurrency adverts being partially ended in October 2018 [63].

The identified significance of US Dollar discussions was consistent with the importance of speculation in the 2017–2018 pricing cycle, an issue raised by the House of Commons Treasury Committee [16]. In

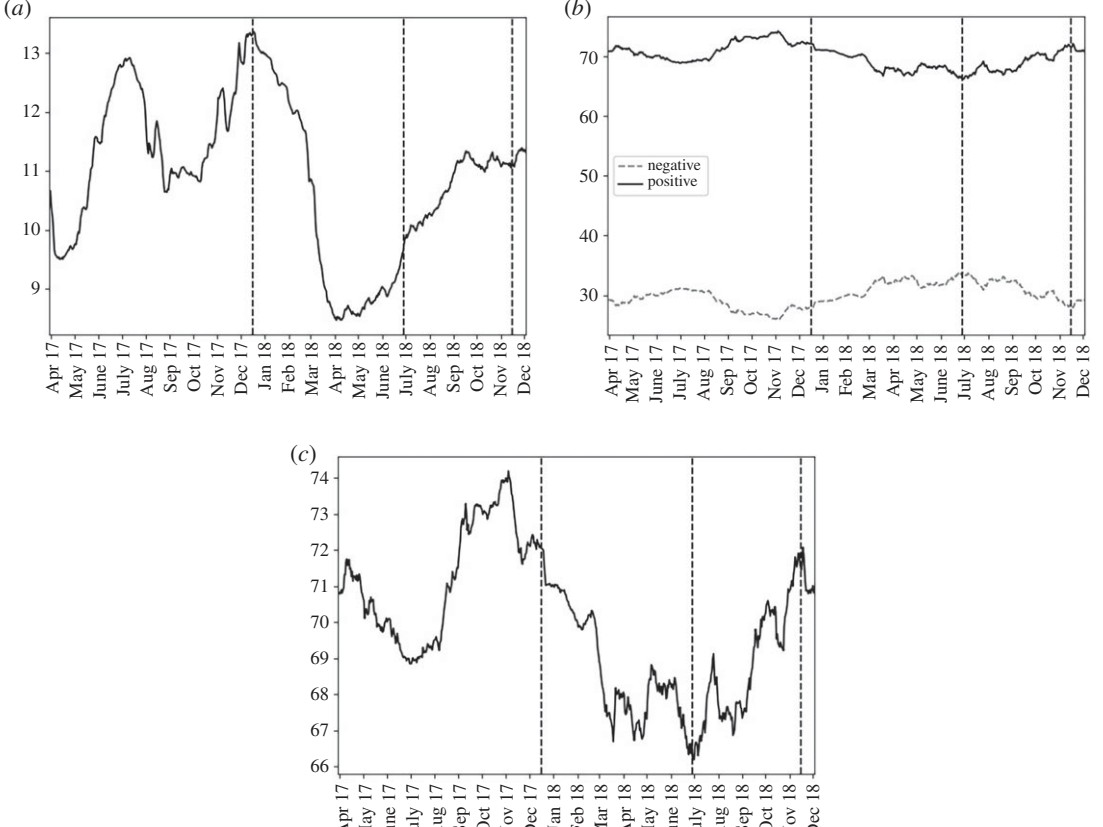

**Figure 7.** For 'dollar_marker_symbol', over the past 90 days, from the top left panel clockwise: (*a*) the percentage of submissions containing the term; (*b*) the percentage of negative and positive submissions that were of positive sentiment (solid, black line) or negative sentiment (dashed, grey line); (*c*) the percentage that were of positive sentiment.

stages 1 and 2, US Dollars were most mentioned in the context of buying bitcoin table 3, with a fall in US Dollar mentions in phase 2 consistent with declining buying enthusiasm figure 7. In the period of relative price stability (phase 3), 'buy' no longer most commonly occurred with US Dollars, nor was it in the chain of popular words table 3. With more stable prices, there was thus less evidence for speculation.

The price dynamic word 'tax' showed a statistically significant increase in frequency from phase 1 to phase 2 and fall from phase 2 to phase 3. 'Tax' most frequently occurred with 'pay' across all phases. The words 'capit[al]' and 'gains' were other close associates in stages 1 and 2. Gains on bitcoin trading have been deemed liable to Capital Gains Tax in the US, UK, Japan and Australia [65]. The price gains in phase 1 would have generated a tax liability for traders who sold bitcoin. In order to meet this, they might have sold further bitcoin in stage 2 when tax was due, thus driving a downwards trend in prices. The positive sentiment figure 8 across all stages may reflect that the need to pay tax is associated with making a financial gain.

# 6. Conclusion

Reddit text provides insight into how human behaviour evolved during the recent, dramatic upswing and downswing in bitcoin prices. Rather than determining whether sentiment is positive or negative [4–7,26,37], we obtain our insights by identifying specific words and their associated context.

We achieve this analysis of Reddit text by developing a new DDPWI methodology that identifies those words that were significantly more or less frequent during a specific time period. We further develop an approach for elucidating the context in which these words were used across different time periods. DDPWI can be applied more generally where there is a need to understand why some temporary social phenomenon occurred. For example, it could explain why a charity had time-limited success in raising donations, what drove the brief craze for a firm's product and the phases in another cryptocurrency's price series.

We apply DDPWI to compare the distinct phase of falling bitcoin prices with before (rising prices) and after (stabilizing prices). This supported an association between bans, tax and speculation and the

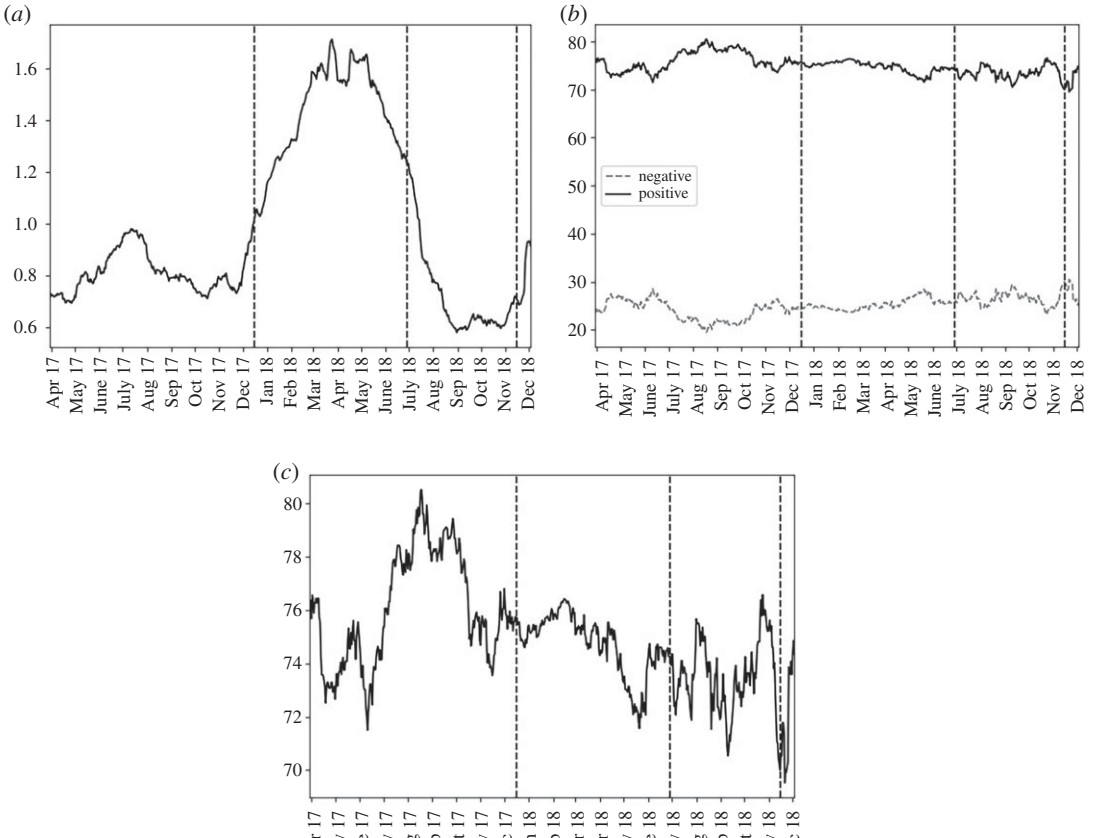

**Figure 8.** For 'tax', over the past 90 days, from the top figure down: (*a*) the percentage of submissions containing the term; (*b*) the percentage of negative and positive submissions that were of positive sentiment (solid, black line) or negative sentiment (dashed, grey line); (*c*) the percentage that were of positive sentiment.

bitcoin price. The importance of bans is inconsistent with bitcoin being advocated as an independent system [14]. It demonstrates the influence of governments and corporations. This suggests a need for investors and participants in bitcoin to consider current and future global regulation and corporate policy in deciding whether to buy, hold or sell the cryptocurrency. It further suggests that improvements in global regulatory certainty over time may help to reduce bitcoin price volatility, rendering it more suitable for its original intended purpose as an international, online payment system [14].

Data accessibility. API sources of data are listed in the text. The data and the code used to prepare and analyse the data are publicly accessible in a Dryad data repository at: https://doi.org/10.5061/dryad.8n6m564.
Authors' contributions. A.B. conducted the data processing and analysis, and drafted the article. E.Y. provided critical feedback on the article and inputted on the data processing and analysis approaches taken. All authors gave final approval for publication and agree to be held accountable for the work performed therein.
Competing interests. We have no competing interests.
Funding. This work was supported by The Alan Turing Institute under the EPSRC grant no. EP/N510129/1 and Turing award number TU/C/000028.

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
