## [Reviewer comments · Royal Society Open Science]

Review History

RSOS-190030.R0 (Original submission)

Review form: Reviewer 1

Is the manuscript scientifically sound in its present form?

No

Are the interpretations and conclusions justified by the results?

No

Is the language acceptable?

No

Is it clear how to access all supporting data?

No

Do you have any ethical concerns with this paper?

No

Have you any concerns about statistical analyses in this paper?

Yes

Recommendation?

Reject

Comments to the Author(s)

Most of the paper is written in passive voice. It would be a more engaging read if it is written in active voice. Some of the wording and explanation in the paper is badly written and needs a thorough revision, e.g.:

- Section 2.1.1. is not clear, needs a rewrite + details.
- Section 2.1.2. is not clear, needs a rewrite + comprehensive details.
- Section 2.2. some parts are almost impossible to understand, especially the brain twister "This compared same-day Reddit and Google, and Reddit and Google with the bitcoin metrics on the same day as well as 1 day and 2 days after."
- Section 2.4. needs a rewrite

Technical feedback:

- Related work does not connect well with the analyses, and the analyses seems cut-off from the results section.
- There is no explanation or statistics of the Reddit dataset.
- Explanation for why certain statistical tests were applied is dearly missed. Please provide background on what each test does to allow the reader to connect to the purpose of the test. Without such explanation the connection between the tests and the data will remain missing.
- What about consideration of other abbreviations of bitcoin, such as "BITC"?
- Figures 2 and 3 should be superimposed (or be one figure) for better comparison between Google and Reddit.
- Perhaps interesting to explore plotting options for Table 1 and 2?
- The separate nature of Figures 4, 5, 6 and 7 make them extremely hard to compare. Perhaps a better idea would be to have three Figures, one each for A, B and C, showing the four terms comparatively and clearly.

Review form: Reviewer 2**Is the manuscript scientifically sound in its present form?**

Yes

Are the interpretations and conclusions justified by the results?

No

Is the language acceptable?

Yes

Is it clear how to access all supporting data?

Yes

Do you have any ethical concerns with this paper?

No

Have you any concerns about statistical analyses in this paper?

No

Recommendation?

Major revision is needed (please make suggestions in comments)

Comments to the Author(s)

The paper presents a method for discovering "words which matter" in online discussions around bitcoin, by focusing on Reddits or /r/Bitcoin. A period of activity is divided into three phases, and words whose usage varies significantly between these phases is identified.

While the statistical analysis seems reasonable, and finds correlations, it is not really clear why this result is important in itself, or how future work can make use of the results. There seems to be a methodological improvement being claimed, in terms of identifying the words that matter from the data rather than pre-judging which words to use. However, the three phases of analysis (Fig 1) are pre-judged, and all that is being found is that there is some words which are used more frequently in some phases than others. How does this result depend on the definition of phases? [Can we discover the boundary dates of the three phases through differences in language usage? If I arbitrarily choose three dates as phase boundaries, will I discover differences in language usage? If so, what does it mean?].

In summary, this seems like a post-hoc analysis informed by recent bitcoin dynamics. I am not sure how others could use either the methodology or the results in other settings. Nor is it clear what new knowledge the results obtained add to existing knowledge on bitcoin dynamics [eg decline in debate concerning segwit, or the increased popularity of words such as investor and market is to be expected. Even if not expected, how can we make use of these results?]

Detailed comments, relating to specific points in the text:

Page 3 - line 3: "role of social media in influencing the price of bitcoin"  This is not supported by the analysis. All that can be said is that there is a correlation between words used and bitcoin price. In fact, a more reasonable interpretation of the results might be that it reveals how users change their talk differently in response to changes in bitcoin prices. This will completely change the tone and significance and positioning and should be considered in the response.

The whole of Section 1.3 seems very contrived to find advantages for Reddit over other sources. If all you are claiming is that Reddit word usage correlates with Bitcoin prices or phases of bitcoins, the usage of Reddit need not be justified. There may be other "better" sources of data, but you have found *one* such, which yields statistically significant results.

Sec 1.3.1 - not clear why wikipedia is viewed as a site "where users input search terms and obtain results". Also, "wikipedia entries can be viewed or edited by any internet user". True, but Reddit comments can also be made by any internet user. So why is this a reason to choose reddit?

Sec 1.3.3 - The number of online users from one sample point on 25 Sep is used to justify that bitcointalk.org is not suitable. This is simply not sufficient.

2.3.3 - Day is chosen based on GMT time zones. Where does the largest volume of discussion come from? If from the USA, and if the talk happens more in the evening US time zones, it may be that you are splitting the largest volume of discussions into two "days", which could affect some of your results. It would be preferable to look at the natural daily increase and decrease in volumes, and choose the most appropriate time zone to split the activity into "days".

Decision letter (RSOS-190030.R0)

25-Feb-2019

Dear Mr Burnie:

Manuscript ID RSOS-190030 entitled "Social media and Bitcoin Metrics: which words matter" which you submitted to Royal Society Open Science, has been reviewed. The comments from reviewers are included at the bottom of this letter.

In view of the criticisms of the reviewers, the manuscript has been rejected in its current form. However, a new manuscript may be submitted which takes into consideration these comments.

Please note that resubmitting your manuscript does not guarantee eventual acceptance, and that your resubmission will be subject to peer review before a decision is made.

Your resubmitted manuscript should be submitted by 25-Aug-2019. If you are unable to submit by this date please contact the Editorial Office.

on behalf of Dr Jon Crowcroft (Associate Editor) and Professor Marta Kwiatkowska (Subject Editor)
openscience@royalsociety.org

Associate Editor Comments to Author (Dr Jon Crowcroft):

Thank you for a submission on an interesting topic - I'm afraid there are some substantial criticisms that the reviewers have made that I agree with - i think with some work, these can be addressed, so for now I'm recommending reject but allowing for a resubmission. The reviews contain technical suggestions for improving the work.

Associate Editor: 2

Comments to the Author:

I think you mean "effect", not "affect" in para 7 in section 4

Neat paper - what do you think the effect of publishing it will be on the effect of sentiment expressed on reddit, upon bitcoin price? (noting that google did the famous study on flu term searches, which accurately tracked a flu epidemic one year, but dismally failed the next year)

Reviewers' Comments to Author:

Reviewer: 1

Comments to the Author(s)

Most of the paper is written in passive voice. It would be a more engaging read if it is written in active voice. Some of the wording and explanation in the paper is badly written and needs a thorough revision, e.g.:

- Section 2.1.1. is not clear, needs a rewrite + details.
- Section 2.1.2. is not clear, needs a rewrite + comprehensive details.
- Section 2.2. some parts are almost impossible to understand, especially the brain twister "This compared same-day Reddit and Google, and Reddit and Google with the bitcoin metrics on the same day as well as 1 day and 2 days after."
- Section 2.4. needs a rewrite

Technical feedback:

- Related work does not connect well with the analyses, and the analyses seems cut-off from the results section.
- There is no explanation or statistics of the Reddit dataset.
- Explanation for why certain statistical tests were applied is dearly missed. Please provide background on what each test does to allow the reader to connect to the purpose of the test. Without such explanation the connection between the tests and the data will remain missing.
- What about consideration of other abbreviations of bitcoin, such as "BITC"?
- Figures 2 and 3 should be superimposed (or be one figure) for better comparison between Google and Reddit.
- Perhaps interesting to explore plotting options for Table 1 and 2?
- The separate nature of Figures 4, 5, 6 and 7 make them extremely hard to compare. Perhaps a better idea would be to have three Figures, one each for A, B and C, showing the four terms comparatively and clearly.

Reviewer: 2

Comments to the Author(s)

The paper presents a method for discovering "words which matter" in online discussions around bitcoin, by focusing on Reddits or /r/Bitcoin. A period of activity is divided into three phases, and words whose usage varies significantly between these phases is identified.

While the statistical analysis seems reasonable, and finds correlations, it is not really clear why this result is important in itself, or how future work can make use of the results. There seems to be a methodological improvement being claimed, in terms of identifying the words that matter from the data rather than pre-judging which words to use. However, the three phases of analysis (Fig 1) are pre-judged, and all that is being found is that there is some words which are used more frequently in some phases than others. How does this result depend on the definition of phases? [Can we discover the boundary dates of the three phases through differences in language usage? If I arbitrarily choose three dates as phase boundaries, will I discover differences in language usage? If so, what does it mean?].

In summary, this seems like a post-hoc analysis informed by recent bitcoin dynamics. I am not sure how others could use either the methodology or the results in other settings. Nor is it clear what new knowledge the results obtained add to existing knowledge on bitcoin dynamics [eg decline in debate concerning segwit, or the increased popularity of words such as investor and market is to be expected. Even if not expected, how can we make use of these results?]

Detailed comments, relating to specific points in the text:

Page 3 - line 3: "role of social media in influencing the price of bitcoin"  This is not supported by the analysis. All that can be said is that there is a correlation between words used and bitcoin price. In fact, a more reasonable interpretation of the results might be that it reveals how users change their talk differently in response to changes in bitcoin prices. This will completely change the tone and significance and positioning and should be considered in the response.

The whole of Section 1.3 seems very contrived to find advantages for Reddit over other sources. If all you are claiming is that Reddit word usage correlates with Bitcoin prices or phases of bitcoins, the usage of Reddit need not be justified. There may be other "better" sources of data, but you have found *one* such, which yields statistically significant results.

Sec 1.3.1 - not clear why wikipedia is viewed as a site "where users input search terms and obtain results". Also, "wikipedia entries can be viewed or edited by any internet user". True, but Reddit comments can also be made by any internet user. So why is this a reason to choose reddit?

Sec 1.3.3 - The number of online users from one sample point on 25 Sep is used to justify that bitcointalk.org is not suitable. This is simply not sufficient.

2.3.3 - Day is chosen based on GMT time zones. Where does the largest volume of discussion come from? If from the USA, and if the talk happens more in the evening US time zones, it may be that you are splitting the largest volume of discussions into two "days", which could affect some of your results. It would be preferable to look at the natural daily increase and decrease in volumes, and choose the most appropriate time zone to split the activity into "days".

Author's Response to Decision Letter for (RSOS-190030.R0)

See Appendix A.

RSOS-190467.R0

Review form: Reviewer 1

Is the manuscript scientifically sound in its present form?

Yes

Are the interpretations and conclusions justified by the results?

Yes

Is the language acceptable?

Yes

Is it clear how to access all supporting data?

No

Do you have any ethical concerns with this paper?

No

Have you any concerns about statistical analyses in this paper?

No

Recommendation?

Major revision is needed (please make suggestions in comments)

Comments to the Author(s)

The language requires consideration for thorough rewrite. There are quite a few paragraphs not able to provide clarity of purpose, analysis or results.

Tables 2 and 3 need to be figures or summarised somehow -- they're not useful in current form.

It is not clear whether code, data, etc. are made available and if so where can they be accessed.

Related Work needs to be expanded.

Review form: Reviewer 2

Is the manuscript scientifically sound in its present form?

Yes

Are the interpretations and conclusions justified by the results?

Yes

Is the language acceptable?

Yes

Is it clear how to access all supporting data?

Yes

Do you have any ethical concerns with this paper?

No

Have you any concerns about statistical analyses in this paper?

No

Recommendation?

Reject

Comments to the Author(s)

With their clarification (Answer 1 to Reviewer 2), it seems like the contributions are twofold:
 1. From the methodology side: the DDPWI methodology is claimed as a contribution. This seems very obvious and basic to this reviewer, and not in itself sufficient to make it publication worthy (I note this is a subjective and personal judgement of what is 'interesting', so debatable).

2. From the application/bitcoin domain perspective: the fact that price dynamic words were found, and an association between bans and viability of bitcoins. I have some reservations about this claim (see below).

Because of the above, I do not believe this is making a significant contribution. It also seems like the authors have not fully understood or internalised previous reviewer comments. I have tried once again to explain, using a more conversational style only the comments I made previously:

Comment 2 ("However the three phases are pre-judged...") has been misunderstood. It seems like the following connection is being made in the paper - phase 1 had price increases, phase 2 had price decreases and phase 3 had stable prices. Any change in word usage across phases is taken as "words that matter" for price. However, this assumes that the definition of phases is somehow correct, and relevant. What if we replaced price with the weather or this reviewer's mood over time? If we find three periods, one when the reviewer was upbeat, one when downbeat and one when neither upbeat nor downbeat, and look at reddit words used, we might conceivably find differences in word usage correlated with other extraneous events happening in the world. It would be specious to claim that these are words which matter for the reviewer's mood. Hence the query in Page 6 line 11 of Comment 2 - "if I arbitrarily chose three dates as phase boundaries, will I discover differences in language usage"

In response to comment 3: No, I was not "equivocating" and do not accept that the "results in themselves are revelatory" (though they may be of interest). The question I was asking was whether I need your new method to find those results. It seems like most results you find should already be known to the community (eg of bitcoin traders). This comment was essentially asking: Why do we need the methodology developed in this paper? What domain problem does it solve?

Comment 4: The authors say "It is unclear why falling prices would drive more discussions of the word 'ban'."  This is the main problem with correlational studies. We may sometimes not find an answer, or if we are very creative may find specious answers. For instance, I can imagine that people on the forum are very distraught that the value of their bitcoin hoard is decreasing and are essentially whining and whinging about it. Hence increased talk of ban when price decreases.

The authors say in the new text, and in comment 4 that: "A more plausible explanation is that it was announcements, implementations or rumours of bans, which resulted in related discussions on social media, that influenced the reductions in price." I agree that bans may have impacted price. And that bans may have impacted social media. I don't see the association bans  social media discussions  influence price. This is wholly unsupported.

Comment 5: Again misunderstood. I am not saying that you need to find "better" sources than Reddit. I am actually saying "hey look - you have found *a* source, which is somehow correlated with bitcoin prices, so just say that, rather than contriving to prove that somehow the choice you had made was great. The current section 1.3 reads OK to me, so this is no longer a concern.

Comment 8: Again misunderstood. I will try to explain with an example: Many of the results depend on daily volumes, and you are splitting days based on GMT. Suppose there were 35 mentions of a word X in Day 1 (GMT), and 26 mentions on the next day (Day 2). If your threshold count is 30 words to be "most significant", then word X will not appear to be used in significant numbers on Day 2 (below the threshold of 30). However, assume you instead used China or India time zones which end before Day 1 (GMT) ends. It may be that in this case, some of the mentions of word X will then be carried over to day 2 beginning. Thus, you may end up with daily counts of 31 and 30 for word X. Both days now show sufficient numbers at or above your threshold of 30 words per day. Your analysis is more sophisticated and robust than my example, but the same boundary issues exist. There needs to be a sensitivity analysis to show for example that the list of

most frequent words (4.4) does not depend on which time zone you use for marking the boundary in this global discussion. Same for other results in Section 4.

Also the first paragraph of your response to comment 8 only says that bitcoin trading is global. That may be so, but Reddit is mostly driven by US and Western Europe, so the volumes you see in the discussion forum may likely be driven by US time zones.

Decision letter (RSOS-190467.R0)

28-May-2019

Dear Mr Burnie:

Manuscript ID RSOS-190467 entitled "Social media and Bitcoin Metrics: which words matter" which you submitted to Royal Society Open Science, has been reviewed. The comments from reviewer(s) are included at the bottom of this letter.

In view of the criticisms of the reviewer(s), I must decline the manuscript for publication in Royal Society Open Science at this time. However, a new manuscript may be submitted which takes into consideration these comments.

Please note that resubmitting your manuscript does not guarantee eventual acceptance, and that your resubmission will be subject to re-review by the reviewer(s) before a decision is rendered.

You will be unable to make your revisions on the originally submitted version of your manuscript. Instead, revise your manuscript using a word processing program and save it on your computer.

You may also click the below link to start the resubmission process (or continue the process if you have already started your resubmission) for your manuscript. If you use the below link you will not be required to login to ScholarOne Manuscripts.

*** PLEASE NOTE: This is a two-step process. After clicking on the link, you will be directed to a webpage to confirm. ***

https://mc.manuscriptcentral.com/rsos?URL_MASK=75c184a6fc7b43febcc7e1526a3233d6

Because we are trying to facilitate timely publication of manuscripts submitted to Royal Society Open Science, your resubmitted manuscript should be submitted by 25-Nov-2019. If you are unable to submit by this date please contact the Editorial Office for options.

I look forward to a resubmission.

Kind regards,
Andrew Dunn

Royal Society Open Science Editorial Office
 Royal Society Open Science
 openscience@royalsociety.org

on behalf of Dr Jon Crowcroft (Associate Editor) and Marta Kwiatkowska (Subject Editor)
 openscience@royalsociety.org

Associate Editor Comments to Author (Dr Jon Crowcroft):

Associate Editor

Comments to the Author:

See reviewer comments, please.

Reviewer comments to Author:

Reviewer: 1

Comments to the Author(s)

The language requires consideration for thorough rewrite. There are quite a few paragraphs not able to provide clarity of purpose, analysis or results.

Tables 2 and 3 need to be figures or summarised somehow -- they're not useful in current form.

It is not clear whether code, data, etc. are made available and if so where can they be accessed.

Related Work needs to be expanded.

Reviewer: 2

Comments to the Author(s)

With their clarification (Answer 1 to Reviewer 2), it seems like the contributions are twofold:

1. From the methodology side: the DDPWI methodology is claimed as a contribution. This seems very obvious and basic to this reviewer, and not in itself sufficient to make it publication worthy (I note this is a subjective and personal judgement of what is 'interesting', so debatable).
2. From the application/bitcoin domain perspective: the fact that price dynamic words were found, and an association between bans and viability of bitcoins. I have some reservations about this claim (see below).

Because of the above, I do not believe this is making a significant contribution. It also seems like the authors have not fully understood or internalised previous reviewer comments. I have tried once again to explain, using a more conversational style only the comments I made previously:

Comment 2 ("However the three phases are pre-judged...") has been misunderstood. It seems like the following connection is being made in the paper - phase 1 had price increases, phase 2 had price decreases and phase 3 had stable prices. Any change in word usage across phases is taken as "words that matter" for price. However, this assumes that the definition of phases is somehow correct, and relevant. What if we replaced price with the weather or this reviewer's mood over time? If we find three periods, one when the reviewer was upbeat, one when downbeat and one when neither upbeat nor downbeat, and look at reddit words used, we might conceivably find differences in word usage correlated with other extraneous events happening in the world. It would be specious to claim that these are words which matter for the reviewer's mood. Hence the query in Page 6 line 11 of Comment 2 - "if I arbitrarily chose three dates as phase boundaries, will I discover differences in language usage"

In response to comment 3: No, I was not "equivocating" and do not accept that the "results in themselves are revelatory" (though they may be of interest). The question I was asking was whether I need your new method to find those results. It seems like most results you find should already be known to the community (eg of bitcoin traders). This comment was essentially asking: Why do we need the methodology developed in this paper? What domain problem does it solve?

Comment 4: The authors say "It is unclear why falling prices would drive more discussions of the word 'ban'."  This is the main problem with correlational studies. We may sometimes not find an answer, or if we are very creative may find specious answers. For instance, I can imagine that people on the forum are very distraught that the value of their bitcoin hoard is decreasing and are essentially whining and whinging about it. Hence increased talk of ban when price decreases.

The authors say in the new text, and in comment 4 that: "A more plausible explanation is that it was announcements, implementations or rumours of bans, which resulted in related discussions on social media, that influenced the reductions in price.". I agree that bans may have impacted price. And that bans may have impacted social media. I dont see the association bans  social media discussions  influence price. This is wholly unsupported.

Comment 5: Again misunderstood. I am not saying that you need to find "better" sources than Reddit. I am actually saying "hey look - you have found *a* source, which is somehow correlated with bitcoin prices, so just say that, rather than contriving to prove that somehow the choice you had made was great. The current section 1.3 reads OK to me, so this is no longer a concern.

Comment 8: Again misunderstood. I will try to explain with an example: Many of the results depend on daily volumes, and you are splitting days based on GMT. Suppose there were 35 mentions of a word X in Day 1 (GMT), and 26 mentions on the next day (Day 2). If your threshold count is 30 words to be "most significant", then word X will not appear to be used in significant numbers on Day 2 (below the threshold of 30). However, assume you instead used China or India time zones which end before Day 1 (GMT) ends. It may be that in this case, some of the mentions of word X will then be carried over to day 2 beginning. Thus, you may end up with daily counts of 31 and 30 for word X. Both days now show sufficient numbers at or above your threshold of 30 words per day. Your analysis is more sophisticated and robust than my example, but the same boundary issues exist. There needs to be a sensitivity analysis to show for example that the list of most frequent words (4.4) does not depend on which time zone you use for marking the boundary in this global discussion. Same for other results in Section 4.

Also the first paragraph of your response to comment 8 only says that bitcoin trading is global. That may be so, but Reddit is mostly driven by US and Western Europe, so the volumes you see in the discussion forum may likely be driven by US time zones.

Author's Response to Decision Letter for (RSOS-190467.R0)

See Appendix B.

RSOS-191068.R0

Review form: Reviewer 1

Is the manuscript scientifically sound in its present form?

Yes

Are the interpretations and conclusions justified by the results?

No

Is the language acceptable?

Yes

Do you have any ethical concerns with this paper?

No

Have you any concerns about statistical analyses in this paper?

No

Recommendation?

Major revision is needed (please make suggestions in comments)

Comments to the Author(s)

Copyright: "2014TheAuthors" 2019?

Abstract: "words that impact on the metric" should be "words that impact the metric"

(Suggestion only) Section 1.1. Perhaps updating the dataset would be worthwhile in getting the latest insights into the paper. This is more important in crypto-asset industry given the volatile nature of the said.

Unclear why Figure 2 is Bitcoin Cash and not Bitcoin? Same comment for Table 2. Shouldn't it be Bitcoin? BTC and BCH aren't the same. As of now, it creates confusion and doesn't create a legitimate comparison. The conclusions drawn in the paper must be justified by all analyses.

Replace tables with figures for Table 3 and Table 4. Visualising the transition from stage to stage will be a value add. This is to justify the conclusions drawn.

Consider publishing the data for further research and repeatability.

Review form: Reviewer 3

Is the manuscript scientifically sound in its present form?

Yes

Are the interpretations and conclusions justified by the results?

Yes

Is the language acceptable?

Yes

Do you have any ethical concerns with this paper?

No

Have you any concerns about statistical analyses in this paper?

No

Recommendation?

Accept with minor revision (please list in comments)

Comments to the Author(s)

This is an interesting and nicely written paper. I have several comments.

- More care should be taken with using bitcoin vs. Bitcoin.
- In the second paragraph of the Introduction section, there are various statements that need some more space or more proper referencing:
 - = The 420 retailers should be somehow time-stamped.
 - = There have many more studies comparing Bitcoin to gold (and other commodities) than Ref. 17 and these should be at least mentioned:
 - == Ji et al. (2019): Information interdependence among energy, cryptocurrencies and major commodity markets, *Energy Economics* 81, pp. 1042-1055
 - == Shahzad et al. (2019): Is Bitcoin a better safe-haven investment than gold and commodities?, *International Review of Financial Analysis* 63, pp. 322-330
 - == Bouri et al. (2017): Bitcoin for energy commodities before and after the December 2013 crash: diversifier, hedge or safe haven? *Applied Economics* 49 (50), pp. 5063-5073
 - = "the value of bitcoin is driven solely by market sentiment" is a very strong statement. There has been research pointing at fundamental drivers of the price, e.g. Refs. [3], [9], [10]. More recently, also:
 - == Kristoufek, L. (2019): Is the Bitcoin price dynamics economically reasonable? Evidence from fundamental laws, *Physica A*, In Press, <https://doi.org/10.1016/j.physa.2019.04.109>
- Section 2.1: It should be more clearly explained why the absolute value of price change is taken as a proxy of volatility as it is known to be considerably biased and there are various much better measures, e.g. realised volatility or range-based estimators if one does not want to engage with high-frequency data.
- Under Eq. 3.1, it is not completely clear whether the ADF tests were run on daily percentage changes or the original data.
- Numbers in Table 2 are very hard to read. I suggest using standard decimal numbers up to some reasonable level, say 4, and then only something like "<0.0001", mostly for p-values.
- There is a huge number of sections and mainly subsections (and sub subsections) that are quite disturbing for a reader. Some restructuring might be helpful.

Decision letter (RSOS-191068.R0)

29-Jul-2019

Dear Mr Burnie,

The Subject Editor assigned to your paper ("Social media and Bitcoin Metrics: which words matter") has now received comments from reviewers. We would like you to revise your paper in

accordance with the referee and Associate Editor suggestions which can be found below (not including confidential reports to the Editor). Please note this decision does not guarantee eventual acceptance.

Please submit a copy of your revised paper before 21-Aug-2019. Please note that the revision deadline will expire at 00.00am on this date. If we do not hear from you within this time then it will be assumed that the paper has been withdrawn. In exceptional circumstances, extensions may be possible if agreed with the Editorial Office in advance. We do not allow multiple rounds of revision so we urge you to make every effort to fully address all of the comments at this stage. If deemed necessary by the Editors, your manuscript will be sent back to one or more of the original reviewers for assessment. If the original reviewers are not available we may invite new reviewers.

When submitting your revised manuscript, you must respond to the comments made by the referees and upload a file "Response to Referees" in "Section 6 - File Upload". Please use this to document how you have responded to each of the comments, and the adjustments you have made. In order to expedite the processing of the revised manuscript, please be as specific as possible in your response.

- Ethics statement

- Data accessibility

<http://datadryad.org/submit?journalID=RSOS&manu=RSOS-191068>

- Competing interests

- Authors' contributions

- Acknowledgements

- Funding statement

on behalf of Marta Kwiatkowska (Subject Editor)
openscience@royalsociety.org

Associate Editor Comments to Author:

Thank you for the resubmission. The reviewers are broadly of the view the paper is approaching publishable status; however, a number of revisions have been requested by the referees - please ensure you respond to these comments in a point-by-point response, as well as incorporating the required changes in the manuscript. It would help if you could ensure a tracked-changes version of the paper was included with the revision.

One point that we'd like to highlight in particular is the reference made by one of the reviewers to data access. Royal Society Open Science requires data, code, or other digital research materials to be accessible throughout peer review and subsequently publication. While we note you've included details in the text and also in the ESM, please do make sure you have made available as much of the dataset etc as possible to allow a reader to attempt a replication of your work, should

they be so inclined. If you're unsure, contact the editorial office (openscience@royalsociety.org) for advice.

Reviewer comments to Author:

Reviewer: 3

Comments to the Author(s)

This is an interesting and nicely written paper. I have several comments.

- More care should be taken with using bitcoin vs. Bitcoin.
- In the second paragraph of the Introduction section, there are various statements that need some more space or more proper referencing:
 - = The 420 retailers should be somehow time-stamped.
 - = There have many more studies comparing Bitcoin to gold (and other commodities) than Ref. 17 and these should be at least mentioned:
 - == Ji et al. (2019): Information interdependence among energy, cryptocurrencies and major commodity markets, *Energy Economics* 81, pp. 1042-1055
 - == Shahzad et al. (2019): Is Bitcoin a better safe-haven investment than gold and commodities?, *International Review of Financial Analysis* 63, pp. 322-330
 - == Bouri et al. (2017): Bitcoin for energy commodities before and after the December 2013 crash: diversifier, hedge or safe haven? *Applied Economics* 49 (50), pp. 5063-5073
 - = "the value of bitcoin is driven solely by market sentiment" is a very strong statement. There has been research pointing at fundamental drivers of the price, e.g. Refs. [3], [9], [10]. More recently, also:
 - == Kristoufek, L. (2019): Is the Bitcoin price dynamics economically reasonable? Evidence from fundamental laws, *Physica A, In Press*, <https://doi.org/10.1016/j.physa.2019.04.109>
- Section 2.1: It should be more clearly explained why the absolute value of price change is taken as a proxy of volatility as it is known to be considerably biased and there are various much better measures, e.g. realised volatility or range-based estimators if one does not want to engage with high-frequency data.
- Under Eq. 3.1, it is not completely clear whether the ADF tests were run on daily percentage changes or the original data.
- Numbers in Table 2 are very hard to read. I suggest using standard decimal numbers up to some reasonable level, say 4, and then only something like "<0.0001", mostly for p-values.
- There is a huge number of sections and mainly subsections (and sub subsections) that are quite disturbing for a reader. Some restructuring might be helpful.

Reviewer: 1

Comments to the Author(s)

Copyright: "2014TheAuthors" 2019?

Abstract: "words that impact on the metric" should be "words that impact the metric"

(Suggestion only) Section 1.1. Perhaps updating the dataset would be worthwhile in getting the latest insights into the paper. This is more important in crypto-asset industry given the volatile nature of the said.

Unclear why Figure 2 is Bitcoin Cash and not Bitcoin? Same comment for Table 2. Shouldn't it be Bitcoin? BTC and BCH aren't the same. As of now, it creates confusion and doesn't create a legitimate comparison. The conclusions drawn in the paper must be justified by all analyses.

Replace tables with figures for Table 3 and Table 4. Visualising the transition from stage to stage will be a value add. This is to justify the conclusions drawn.

Consider publishing the data for further research and repeatability.

Author's Response to Decision Letter for (RSOS-191068.R0)

See Appendix C.

RSOS-191068.R1 (Revision)

Review form: Reviewer 1

Is the manuscript scientifically sound in its present form?

Yes

Are the interpretations and conclusions justified by the results?

Yes

Is the language acceptable?

Yes

Do you have any ethical concerns with this paper?

No

Have you any concerns about statistical analyses in this paper?

No

Recommendation?

Accept as is

Comments to the Author(s)

Well done, reads and is much better now!

Review form: Reviewer 3

Is the manuscript scientifically sound in its present form?

Yes

Are the interpretations and conclusions justified by the results?

Yes

Is the language acceptable?

Yes

Do you have any ethical concerns with this paper?

No

Have you any concerns about statistical analyses in this paper?

No

Recommendation?

Accept as is

Comments to the Author(s)

All my comments have been sufficiently reflected on.

Decision letter (RSOS-191068.R1)

18-Sep-2019

Dear Mr Burnie,

I am pleased to inform you that your manuscript entitled "Social media and Bitcoin Metrics: which words matter" is now accepted for publication in Royal Society Open Science.

Kind regards,

Andrew Dunn

on behalf of Prof Marta Kwiatkowska (Subject Editor)

Associate Editor Comments to Author:

This latest version of your paper has been reviewed, and we're delighted to communicate to you that the paper can now be accepted - congratulations!

Reviewer comments to Author:

Reviewer: 3

Comments to the Author(s)

All my comments have been sufficiently reflected on.

Reviewer: 1

Comments to the Author(s)

Well done, reads and is much better now!

Appendix A

Associate Editor: Dr Jon Crowcroft

Thank you for a submission on an interesting topic - I'm afraid there are some substantial criticisms that the reviewers have made that I agree with - i think with some work, these can be addressed, so for now I'm recommending reject but allowing for a resubmission. The reviews contain technical suggestions for improving the work.

Thank you very much for your in-depth reviews of the paper. We have extensively revised the paper to take into account the reviewers' comments and made a number of amendments to address each of the criticisms as itemised below. We believe that this has substantially improved the paper.

Associate Editor: 2

I think you mean "effect", not "affect" in para 7 in section 4

This has been amended in the revised submission.

Neat paper - what do you think the effect of publishing it will be on the effect of sentiment expressed on reddit, upon bitcoin price? (noting that google did the famous study on flu term searches, which accurately tracked a flu epidemic one year, but dismally failed the next year)

This is a very interesting consideration. We anticipate that the main effect of publication would be an increased number of people looking at the Reddit discussion forum and participating in the discussions. The effect of sentiment expressed upon the bitcoin price is more likely to depend on the nature of the topic under discussion than the number of people participating in the discussion – but the advantage of a larger number of participants is that a more balanced, greater range of views is likely to be presented.

The Google Flu Trends project underlines the limitations of Google search volumes, because the context motivating different searches is unknown. For example, a suggested problem has been that Google searches for flu or flu symptoms (e.g. "fever", "cough") might be made by people with flu-like symptoms who did not actually have the flu. Another explanation was, if flu becomes a prominent news item, this might skew results because people searched for news on flu while not actually having the flu. As a consequence, after some initial success, the Google Trends data was found to consistently overestimate flu prevalence.

Reviewer: 1

Thank you for the specific details on which sections and figures should be revised.

1. Most of the paper is written in passive voice. It would be a more engaging read if it is written in active voice.

The active voice has been adopted in the revised paper where appropriate.

2. Some of the wording and explanation in the paper is badly written and needs a thorough revision, e.g.:

- Section 2.1.1. is not clear, needs a rewrite + details.

Section 2.1.1 becomes Section 2.2 in revised version.

In the revised paper, extensive detail on how Google search volume data were collected and rescaled is now provided.

- Section 2.1.2. is not clear, needs a rewrite + comprehensive details.

Section 2.1.2 (2.3 in revised version) has been altered for clarity. Comprehensive details are provided in Section 2.4, which covers how submissions were filtered (2.4.1), how text was processed (2.4.2) and how daily frequencies were calculated (2.4.3).

- Section 2.2. some parts are almost impossible to understand, especially the brain twister "This compared same-day Reddit and Google, and Reddit and Google with the bitcoin metrics on the same day as well as 1 day and 2 days after."

Section 2.2 (now 3.1) has been restructured with the above quote replaced with the following (3rd paragraph of Section 3.1):

"Correlation analyses were applied to evaluate the similarities between Reddit and Google. This involved calculating correlations between Reddit submissions (Section 2.3.) and Google search volumes (Section 2.2.), and between these and the ten Bitcoin metrics (Section 2.1.). In addition, we assessed how Reddit submissions and Google searches were associated with the values of the Bitcoin metrics recorded 1 and 2 days after the Reddit and Google volume metrics."

- Section 2.4. needs a rewrite

Section 2.4 has been split into two sections in the revised paper: identifying important words using an absolute frequency cut-off of 5% (Section 3.2); and identifying important words through comparing their relative daily frequencies across stages (Section 3.3). Section 3.3 details both the statistical approach for comparing stages (Section 3.3.1) and how this was used in determining the 'price dynamic words' using a new 'Data-Driven Phasic Word

Identification' methodology (Section 3.3.2). The iterative procedure by which the context of these price dynamic words is uncovered is described in Section 3.4.1.

3. Technical feedback

- a. Related work does not connect well with the analyses, and the analyses seems cut-off from the results section.

These issues have been addressed in the new version.

The section on contributions of this article (Section 1.3) has been revised to better link the related work (Section 1.2) with the analyses described in the new methodology (Section 3).

The analyses (Section 3 Methodology) have been separated from Data Preparation (Section 2) for added clarity. The Results (Section 4) follows the Methodology.

Greater detail is provided on explanations as to the reasoning behind the selection of the statistical analyses performed in Section 3. These analyses implement the non-parametric approach advocated in the Contributions (Section 1.3).

Within the Methodology (Section 3), the analyses are now more clearly differentiated. The new different titled subsections can now be more easily related to similarly titled subsections in the Results (Section 4).

- b. There is no explanation or statistics of the Reddit dataset.

An explanation for the choice of Reddit is provided in Section 1.4 (Why choose Reddit).

Descriptive statistics are now provided in a new Section 4.1 and Table 1:

4.1. **Reddit Submissions Descriptive Statistics**

Table 1 presents descriptive statistics on Reddit submissions, showing a decline in Reddit activity as prices stabilised. On average, over 500 submissions were posted per day when prices were most volatile in stages 1 and 2; this fell 46% with stage 3.

Table 1. Descriptive statistics for Reddit Submissions for across the dataset (1 January 2017 to 3 December 2018) and within stages 1, 2 and 3 (see Section 1.1.).

Stage	Days	Submissions	Submissions per Day
All Data	702	326945	465.73
1	349	181327	519.56
2	195	101110	518.51
3	139	38706	278.46

- c. Explanation for why certain statistical tests were applied is clearly missed. Please provide background on what each test does to allow the reader to connect to the purpose of the test. Without such explanation the connection between the tests and the data will remain missing.

The Methodology (Section 3) has been revised to clarify the reasoning behind the statistical approach followed. The Related Work (Section 1.2) discusses the limitations in using linear regressions and wavelet analysis, which led us to apply non-parametric approaches that make minimal assumptions regarding the price relationship and the underlying distributions. This justified choosing Spearman's rho over Pearson Product-Moment correlation (Section 3.1), and Wilcoxon Rank-Sum Test over the t-test (Section 3.3.1).

- d. What about consideration of other abbreviations of bitcoin, such as "BITC"?

We converted 'btc' and 'xbt', which are the accepted currency codes for bitcoin (<https://support.kraken.com/hc/en-us/articles/360001206766-Bitcoin-currency-code-XBT-vs-BTC>), into the synonymous 'bitcoin'. In the revised version (Section 2.4.2) we have added the reference that justifies this.

"BITC" is the name of a Swiss Bitcoin exchange (<https://bitc.ch/>).

- e. Figures 2 and 3 should be superimposed (or be one figure) for better comparison between Google and Reddit. Perhaps interesting to explore plotting options for Table 1 and 2? The separate nature of Figures 4, 5, 6 and 7 make them extremely hard to compare. Perhaps a better idea would be to have three Figures, one each for A, B and C, showing the four terms comparatively and clearly.

Alternative approaches to result presentation were considered in line with suggestions.

Superimposing Figures 2 and 3 worked well in enhancing comparability (see new Figure 2).

Regarding Tables 1 and 2 (Tables 2 and 3 in the revised version), other plotting options were considered but they reduced clarity. The method of presentation was in line with how linear regression results have been presented in previous literature (see referenced papers in Section 1.2).

Having three Figures, one each for A, B and C, rather than four was considered but had the problem that different words were of different scales for each of these parameters, so that combining them would reduce resolution and clarity.

Reviewer: 2

The paper presents a method for discovering "words which matter" in online discussions around bitcoin, by focusing on Reddits or /r/Bitcoin. A period of activity is divided into three phases, and words whose usage varies significantly between these phases is identified.

1. While the statistical analysis seems reasonable, and finds correlations, it is not really clear
 - a. why this result is important in itself, or how future work can make use of the results.
 - b. There seems to be a methodological improvement being claimed, in terms of identifying the words that matter from the data rather than pre-judging which words to use.

The Conclusion has been split from the Discussion to form a new Section 6 which summarises the importance of the results obtained and how the new methodology could be applied in future work.

- a. There is a contribution in terms of the new results being based on recent data (years 2017-18). These are detailed in the Results (Section 4) and Discussion (Section 5), and the importance of these results is highlighted through discussing their implications in the Conclusion (Section 6). In particular, please note in Section 6, 3rd paragraph:

"This supported an association between bans, tax and speculation and the bitcoin price. The importance of bans is inconsistent with bitcoin being advocated as an independent system [14]. It demonstrates the influence of governments and corporations. This suggests a need for investors and participants in bitcoin to consider current and future global regulation and corporate policy in deciding whether to buy, hold or sell the cryptocurrency. It further suggests that improvements in global regulatory certainty over time may help to reduce bitcoin price volatility, rendering it more suitable for its original intended purpose as an international, online payment system [14]."

- b. A new Data-Driven Phasic Word Identification methodology is developed to recognise words that differed in frequency during a specific time phase compared with before and after this time phase. This contribution is now more clearly emphasised in the abstract, Section 1.3, in the methodology (Section 3.3.2) and in the Conclusion (Section 6).

The reviewer is correct. With this new methodological approach, identification of words is data-driven, avoiding the need to pre-judge which words matter.

The methodology is broadly applicable to settings where there is a need to understand why some temporary social phenomenon occurred (see Section 6). It could be applied

to understand why a charity had time-limited success in raising donations, what drove the brief craze for a firm's product and to other cryptocurrency price series.

2. However, the three phases of analysis (Fig 1) are pre-judged, and all that is being found is that there is some words which are used more frequently in some phases than others. How does this result depend on the definition of phases? [Can we discover the boundary dates of the three phases through differences in language usage? If I arbitrarily choose three dates as phase boundaries, will I discover differences in language usage? If so, what does it mean?].

The three phases were selected objectively (Figure 1) and not 'pre-judged'. They are defined by the three different price dynamics observed and described in Section 1.1. The boundary between stages 1 and 2 is 16 December 2017 because this is when prices reached an unprecedented peak, having risen twenty-fold since 1 January 2017. Thereafter prices fell to 30% of this peak (stage 2). The boundary between stages 2 and 3 is 29 June 2018, because after this date prices stopped falling, consistently remaining above the 29 June price until 15 November, within a relatively tight price band (at 30-42% of the December peak value).

The analysis shows that certain price dynamic words were of a higher or lower frequency in stage 2 (falling prices) than before (rising prices, stage 1) and after (stabilising prices, stage 3).

The language usage on Reddit evolves over time. Establishing boundary dates based on 'differences in language use' would require delineating important from unimportant evolutions in language use. This would require subjectively pre-judging what made a given observed change in language important.

We circumvent this problem through identifying the distinct phases observable in the price series prior to carrying out our analysis on comparing word usage within these phases. The changes in language that coincided with these shifts are more relevant to understanding why the price dynamics changed compared with changes in language across arbitrary time periods. The arbitrary selection of three dates as phase boundaries lacks meaningful interpretation as there is no dynamic such as price to evaluate significance.

3. In summary, this seems like a post-hoc analysis informed by recent bitcoin dynamics. I am not sure how others could use either the methodology or the results in other settings. Nor is it clear what new knowledge the results obtained add to existing knowledge on bitcoin dynamics [eg decline in debate concerning segwit, or the increased popularity of words such as investor and market is to be expected. Even if not expected, how can we make use of these results?]

The Conclusion has been split from the Discussion to form a new Section 6. The new Conclusion details both how the new methodology might be used elsewhere and why the results (when DDPWI is applied to bitcoin) provide important insights into the impact of government and corporate bans on bitcoin price and volatility. What new knowledge the results obtained add to existing knowledge is detailed in the Results Section 4 and reviewed in the Discussion Section 5.

The results in themselves are revelatory and thus of interest. The reviewer accepts this point in equivocating as to whether the "decline in debate concerning segwit ... is to be expected" or "not expected". The reviewer does not comment on the 'price dynamic words' even though these were of the greatest interest.

4. Detailed comments, relating to specific points in the text:

Page 3 - line 3: "role of social media in influencing the price of bitcoin"  This is not supported by the analysis. All that can be said is that there is a correlation between words used and bitcoin price. In fact, a more reasonable interpretation of the results might be that it reveals how users change their talk differently in response to changes in bitcoin prices. This will completely change the tone and significance and positioning and should be considered in the response.

The referenced text is a comment in the Introduction. It has been altered to 'It therefore characterizes how internet and social media behaviour varied with changes in the price of bitcoin' (2nd paragraph of Section 1.3).

It is unclear why falling prices would drive more discussions of the word 'ban'. See Section 5, 5th paragraph:

"The decision of whether or not to implement a ban through government regulation or corporate policy is an exogenous event not directly dependent on changes in the bitcoin price, thus the higher frequency of 'ban' as prices fell was not likely to be caused by the falling prices. A more plausible explanation is that it was announcements, implementations or rumours of bans, which resulted in related discussions on social media, that influenced the reductions in price."

5. The whole of Section 1.3 seems very contrived to find advantages for Reddit over other sources. If all you are claiming is that Reddit word usage correlates with Bitcoin prices or phases of bitcoins, the usage of Reddit need not be justified. There may be other "better" sources of data, but you have found *one* such, which yields statistically significant results.

We extensively discuss an objective case for using Reddit over other sources in Section 1.4 (1.3 in the original paper). The reviewer does not explain by what criteria other sources might be deemed "better".

Comparison with Google search volumes further supports the relevance of Reddit submissions (methodology in Section 3.1 and results in Sections 4.2 and 4.3).

The reviewer is correct: we do show statistically significant changes in frequency in Reddit word usage in different phases of the bitcoin price series.

6. Sec 1.3.1 - not clear why wikipedia is viewed as a site "where users input search terms and obtain results". Also, "wikipedia entries can be viewed or edited by any internet user". True, but Reddit comments can also be made by any internet user. So why is this a reason to choose reddit?

This text has been changed (see Section 1.4.1) to the following for clarity: "Wikipedia entries can be viewed or edited by any internet user with the previous version lost, inhibiting tracking changes in opinions over time [32]." This is not an issue for discussion forums such as Reddit, as the previous posts are not lost when new posts are made.

7. Sec 1.3.3 - The number of online users from one sample point on 25 Sep is used to justify that bitcointalk.org is not suitable. This is simply not sufficient.

The original paper examined two sample points on 25 September. Re-running the comparison (19:45, 5 March 2019; GMT) showed that the Reddit subreddit continues to have more than four times as many online users. This excludes 330 million-plus monthly active users who could also potentially examine this subreddit.

The bitcointalk.org guidelines do not require online users to focus on bitcoin, unlike the subreddit discussion forum. Consequently, bitcointalk.org includes irrelevant posts which exaggerate interest and reduce the quality of the dataset. In contrast, the Bitcoin subreddit's guidelines clearly state that the 'primary topic is Bitcoin' (Section 1.4.3).

We cannot see an objective rationale for preferring bitcointalk.org over Reddit.

8. 2.3.3 - Day is chosen based on GMT time zones. Where does the largest volume of discussion come from? If from the USA, and if the talk happens more in the evening US time zones, it may be that you are splitting the largest volume of discussions into two "days", which could affect some of your results. It would be preferable to look at the natural daily increase and decrease in volumes, and choose the most appropriate time zone to split the activity into "days".

Day is chosen based on a GMT time-zone because this is the time-zone used by Blockchain Luxembourg S.A. in providing daily bitcoin metrics. Changing the time-zone would result in a mis-match between the bitcoin metrics (particularly price) and the word frequency data invalidating the analysis.

Bitcoin trading is not US-centric and this is reflected in trading statistics (see reference <https://www.bankingtech.com/2018/08/infographic-which-country-trades-the-most-bitcoin/>). Of the total trading volume of bitcoin, 22.77% occurs in the USA (GMT – 4 to – 10) compared with over 25% on a Europe/Africa time-zone (GMT to GMT + 3), 16.55% in Russia (GMT + 2 to + 12) and over 15% on an Asian-Pacific time-zone (GMT + 7 to GMT + 11).

Therefore, GMT is selected as the most central time-zone. Events from across the world could potentially influence social media discussions and price. These could occur in South Korea, China and India (Section 5, 6th paragraph) as well as in the USA and Europe.

Appendix B

Reviewer: 1

Comments to the Author(s)

The language requires consideration for thorough rewrite. There are quite a few paragraphs not able to provide clarity of purpose, analysis or results.

In the amended version, the purpose of the research is now stated at the end of the Introduction:

'We examine the specific research problem of determining what was being discussed on social media during the phase of falling prices compared with the phases before (rising prices) and after (stable prices). This requires a new methodology to delineate significant words (Section 3.3.2.) and the context in which they are being used (Section 3.4.).'

In addition, the paper has been extensively re-written in response to reviewers' comments, to improve clarity of purpose, analysis and results.

Tables 2 and 3 need to be figures or summarised somehow -- they're not useful in current form.

We believed that Tables 2 and 3 were useful in showing all the results, but in response to the reviewer's request, we have merged and summarised them to produce a new table (the new Table 2) which focusses on T=0 results as these were the majority of the statistically significant correlates. The corresponding text in Section 4.3 has also been updated to reflect this new focus, with the results for T=1 and T=2 now summarised as follows: 'Correlating internet activity with the bitcoin metric values one and two days after resulted in findings whose statistical significance was not robust to dividing the data into stages'.

It is not clear whether code, data, etc. are made available and if so where can they be accessed.

Where the data can be accessed is given: for Bitcoin metrics in Section 2.1, for Google search volumes in Section 2.2 and for Reddit submissions text in Section 2.3. Code that details how the Reddit text data were extracted from the Pushshift API and processed was made available by being uploaded as supplementary material. This has been re-uploaded.

Related Work needs to be expanded.

The Related Work section reviewed 23 publications. In the revised version this has been expanded to reflect new relevant publications from the last 6 months.

Reviewer: 2

Comments to the Author(s)

With their clarification (Answer 1 to Reviewer 2), it seems like the contributions are twofold:

1. From the methodology side: the DDPWI methodology is claimed as a contribution. This seems very obvious and basic to this reviewer, and not in itself sufficient to make it publication worthy (I note this is a subjective and personal judgement of what is 'interesting', so debatable).
2. From the application/bitcoin domain perspective: the fact that price dynamic words were found, and an association between bans and viability of bitcoins. I have some reservations about this claim (see below).

The contributions are clearly stated in the paper, in Section 1.3, 'Contributions of this Article':

1. Demonstration that Reddit submissions and Google search volumes behave in a similar manner.
2. New data derived from analysis of the recent 2017-2018 bitcoin coin pricing cycle, which can be divided into 3 phases.
3. Development of a new data-driven phasic word identification methodology (DDPWI) whereby we identify 'price dynamic words' where the change in frequency from phase 1 to 2 (rising prices shifting to falling) and phase 2 to 3 (falling prices ceasing to fall further) are opposite and statistically significant.
4. Development of a novel iterative procedure to identify the context of the use of price dynamic words (Section 3.4.1) and examine the VADER sentiment of submissions containing these words (Section 3.4.2), so as to assist in result interpretation.
5. Shift to non-parametric approaches: the new DDPWI methodology uses the Wilcoxon Rank-Sum Test (not t-tests); Reddit and Google are compared using Spearman's rho (not Pearson Product-Moment Correlation or linear regression). These non-parametric tests reduce sensitivity to extreme outliers and avoids the assumption that the relationship between variables is linear.
6. Shift from testing pre-selected drivers of price to identifying price-sensitive words. This article moves 'the progression of analyses from considering the volume of activity [1,3], to the sentiment of activity [2,7,32] to the actual words used'.

Regarding specifically the DDPWI contribution: a sound justification of the approach may make it seem a natural next step ('obvious') and a clear explanation of the methodology may make it straightforward to implement ('basic'), but, having examined the wider literature, nothing like DDPWI has been applied previously – previous literature has used linear regression or wavelet analysis, based on the false assumptions that bitcoin price distribution is normal and extreme outliers unlikely, which is why instead we used non-parametric tests. Like many other scientific advances each individual part may not be novel but the sum of the parts working in tandem and their ability to tackle a specific problem (here, how to identify significant changes in social media discussions) is the power of the invention. On this objective basis, this research is neither 'obvious' nor 'basic'.

Regarding specifically the ‘application/bitcoin domain perspective’: the reviewer says they have ‘some reservations’ about the claim of ‘an association between bans and viability of bitcoins’. No such association is claimed. Nor do we make any comment on the ‘viability’ of bitcoin.

Because of the above, I do not believe this is making a significant contribution. It also seems like the authors have not fully understood or internalised previous reviewer comments. I have tried once again to explain, using a more conversational style only the comments I made previously:

We have summarised the contributions under Section 1.3 (‘Contributions of this Article’), which include a new dataset, a new methodology and new results from 2017-18. The publication is being delayed due to, in the reviewer’s own words “a subjective and personal judgement of what is ‘interesting’, so debatable” and an incorrect claim that we find “an association between bans and viability of bitcoins”.

Comment 2 (“However the three phases are pre-judged...”) has been misunderstood. It seems like the following connection is being made in the paper - phase 1 had price increases, phase 2 had price decreases and phase 3 had stable prices. Any change in word usage across phases is taken as “words that matter” for price. However, this assumes that the definition of phases is somehow correct, and relevant. What if we replaced price with the weather or this reviewer's mood over time? If we find three periods, one when the reviewer was upbeat, one when downbeat and one when neither upbeat nor downbeat, and look at reddit words used, we might conceivably find differences in word usage correlated with other extraneous events happening in the world. It would be specious to claim that these are words which matter for the reviewer's mood. Hence the query in Page 6 line 11 of Comment 2 - “if I arbitrarily chose three dates as phase boundaries, will I discover differences in language usage”

We do not claim that ‘any change in word usage across phases is taken as “words that matter” for price’; please read Section 3.3.2: ‘the DDPWI approach identifies those words where the change in frequency from phase 1 to 2 (rising prices shifting to falling) and from phase 2 to 3 (falling prices ceasing to fall further) are opposite and both statistically significant’.

Bitcoin is a financial asset currently worth around \$140 billion (coinmarketcap.com). Like other financial assets of that size (e.g. equity or bonds), the idea that ‘if we replaced price with the weather or this reviewer's mood over time ... we might conceivably find differences in word usage’ is irrelevant as financial assets are assessed primarily by value. We are interested in the association between social media discussions and the bitcoin price; not how discussions changed with the weather or the reviewer’s mood.

There seems to be some confusion regarding basic statistical theory. The DDPWI methodology uses Wilcoxon Rank-Sum testing (see Section 3.3) to reduce the likelihood of words being delineated where the observed changes in daily word frequency are ‘specious’ in being attributable purely to random

variation. Statistical tests can conceivably produce false positives, that is why we use a particularly stringent Bonferroni-corrected 1% p-value threshold to reduce this risk. The statistical evidence supports an association between bans and price.

In response to comment 3: No, I was not "equivocating" and do not accept that the "results in themselves are revelatory" (though they may be of interest). The question I was asking was whether I need your new method to find those results. It seems like most results you find should already be known to the community (eg of bitcoin traders). This comment was essentially asking: Why do we need the methodology developed in this paper? What domain problem does it solve?

The reviewer concedes that the results revealed 'may be of interest' – which is consistent with the paper being publishable. You do need the new method to find the results - without it you would not be able to identify which words change significantly in frequency from phase 1 to phase 2 and, in the opposite direction, from phase 2 to 3 versus (1) frequently used words (Section 4.4) and (2) words that demonstrate a statistically significant change in frequency when moving between phases but not in the opposite direction (Section 4.5).

The suggestion that most of the results would already be known to the bitcoin community is unsubstantiated. It assumes that they not only read all the Reddit posts on bitcoin but can simultaneously assimilate significant changes in word usage against changes in price. It does not address the obvious issue that they are exposed to a smorgasbord of information and need a method, such as the one described here, to decide which words matter.

Comment 4: The authors say "It is unclear why falling prices would drive more discussions of the word 'ban'."  This is the main problem with correlational studies. We may sometimes not find an answer, or if we are very creative may find specious answers. For instance, I can imagine that people on the forum are very distraught that the value of their bitcoin hoard is decreasing and are essentially whining and whinging about it. Hence increased talk of ban when price decreases.

The authors say in the new text, and in comment 4 that: "A more plausible explanation is that it was announcements, implementations or rumours of bans, which resulted in related discussions on social media, that influenced the reductions in price.". I agree that bans may have impacted price. And that bans may have impacted social media. I don't see the association bans  social media discussions  influence price. This is wholly unsupported.

The DDPWI method flagged discussion of bans as negatively associated with the bitcoin price. The reviewer speculates that falling prices could result in more discussion of bans. The reasoning behind this assertion is that 'whining and whinging' would include more talk of bans. This is left as a wholly unsupported supposition.

If falling prices result in talk of bans and if 'bans may have impacted price' then we would have expected that the commencement of bans in September – November 2017 (see Section 5) to have led

to a negative feedback loop. Talk of bans reduce prices, falling prices increase talk of bans, and so prices fall further. Why then did bitcoin prices stop falling in phase 3?

To reiterate Section 5, 'a more plausible explanation is that it was announcements, implementations or rumours of bans, which resulted in related discussions on social media, that influenced the reductions in price' rather than vice-versa. In this scenario, prices stopped falling in phase 3 because there were no more new bans being rumoured or implemented.

Comment 5: Again misunderstood. I am not saying that you need to find "better" sources than Reddit. I am actually saying "hey look - you have found *a* source, which is somehow correlated with bitcoin prices, so just say that, rather than contriving to prove that somehow the choice you had made was great. The current section 1.3 reads OK to me, so this is no longer a concern.

Good.

Comment 8: Again misunderstood. I will try to explain with an example: Many of the results depend on daily volumes, and you are splitting days based on GMT. Suppose there were 35 mentions of a word X in Day 1 (GMT), and 26 mentions on the next day (Day 2). If your threshold count is 30 words to be "most significant", then word X will not appear to be used in significant numbers on Day 2 (below the threshold of 30). However, assume you instead used China or India time zones which end before Day 1 (GMT) ends. It may be that in this case, some of the mentions of word X will then be carried over to day 2 beginning. Thus, you may end up with daily counts of 31 and 30 for word X. Both days now show sufficient numbers at or above your threshold of 30 words per day. Your analysis is more sophisticated and robust than my example, but the same boundary issues exist. There needs to be a sensitivity analysis to show for example that the list of most frequent words (4.4) does not depend on which time zone you use for marking the boundary in this global discussion. Same for other results in Section 4.

Also the first paragraph of your response to comment 8 only says that bitcoin trading is global. That may be so, but Reddit is mostly driven by US and Western Europe, so the volumes you see in the discussion forum may likely be driven by US time zones.

The reviewer's suggestion is that we should shift the time-zone used for the daily Reddit word frequency data so that it no longer matches the time-zone used for the price data. As previously stated, the day is chosen on a GMT time-zone because this is the time-zone used by Blockchain Luxembourg S.A. in providing daily bitcoin metrics. Therefore, the sensitivity analysis suggested is fundamentally flawed because the 24-hour time-zone for the price is no longer the same as the 24-hour time-zone of the discussion.

Appendix C

Associate Editor Comments to Author:

Thank you for the resubmission. The reviewers are broadly of the view the paper is approaching publishable status; however, a number of revisions have been requested by the referees - please ensure you respond to these comments in a point-by-point response, as well as incorporating the required changes in the manuscript. It would help if you could ensure a tracked-changes version of the paper was included with the revision.

One point that we'd like to highlight in particular is the reference made by one of the reviewers to data access. Royal Society Open Science requires data, code, or other digital research materials to be accessible throughout peer review and subsequently publication. While we note you've included details in the text and also in the ESM, please do make sure you have made available as much of the dataset etc as possible to allow a reader to attempt a replication of your work, should they be so inclined. If you're unsure, contact the editorial office (openscience@royalsociety.org) for advice.

Thank you for the feedback which we have incorporated into the paper. Our response to these comments is detailed below.

A tracked-changes version ('trackedChanges.pdf') has been included with the revised article.

The dataset and code used have been uploaded to the Dryad depository. Please see: <https://datadryad.org/review?doi=doi:10.5061/dryad.8n6m564>.

This is stated in the amended Data Accessibility Statement and at the beginning of Section 2.

Reviewer comments to Author:

Reviewer: 3

Comments to the Author(s)

This is an interesting and nicely written paper. I have several comments.

- More care should be taken with using bitcoin vs. Bitcoin.

We have opted for 'bitcoin' throughout and have made corrections where this was not the case.

- In the second paragraph of the Introduction section, there are various statements that need some more space or more proper referencing:

= The 420 retailers should be somehow time-stamped.

The 420 retailers have been time-stamped in the revised version.

= There have many more studies comparing Bitcoin to gold (and other commodities) than Ref. 17 and these should be at least mentioned:

== Ji et al. (2019): Information interdependence among energy, cryptocurrencies and major commodity markets, *Energy Economics* 81, pp. 1042-1055

== Shahzad et al. (2019): Is Bitcoin a better safe-haven investment than gold and commodities?, *International Review of Financial Analysis* 63, pp. 322-330

== Bouri et al. (2017): Bitcoin for energy commodities before and after the December 2013 crash: diversifier, hedge or safe haven? *Applied Economics* 49 (50), pp. 5063-5073

= "the value of bitcoin is driven solely by market sentiment" is a very strong statement. There has been research pointing at fundamental drivers of the price, e.g. Refs. [3], [9], [10]. More recently, also:

== Kristoufek, L. (2019): Is the Bitcoin price dynamics economically reasonable? Evidence from fundamental laws, *Physica A*, In

Press, <https://eur01.safelinks.protection.outlook.com/?url=https%3A%2F%2Fdoi.org%2F10.1016%2Fj.physa.2019.04.109&data=02%7C01%7Ccaburnie%40turing.ac.uk%7Cdf1946248c34e78685408d714082f36%7C4395f4a7e4554f958a9f1fbaef6384f9%7C0%7C0%7C636999897698844158&data=1wLuvpS5re%2BUlyusnwYCL11MMhY4D%2BjXQgWG9%2F28v4%3D&reserved=0>

Paragraph 2 of Section 1 ('Introduction') has been revised to better clarify the reasoning behind the importance of market sentiment and incorporates all of the new references suggested.

Shahzad et al (2019) has been added as an additional reference comparing bitcoin with gold.

Ji et al. (2019) and Bouri et al. (2017) show bitcoin prices are weakly connected with energy prices, which supports the concept that the demand for bitcoin as an investment is more price relevant than the cost of supply. When investors buy bitcoin they do so because they believe bitcoin is a good investment. Hence, belief in bitcoin, or market sentiment, impacts bitcoin price.

Kristoufek (2019) links the bitcoin price to transaction data finding that transaction data can explain 88% of price variation. The importance of transactions is consistent with the importance of market

sentiment, with changes in opinion leading to bitcoin being bought or sold, resulting in changes to transaction data.

We have changed the language to the following: 'the value of bitcoin is impacted by market sentiment regarding whether bitcoin is perceived as a good investment'.

- Section 2.1: It should be more clearly explained why the absolute value of price change is taken as a proxy of volatility as it is known to be considerably biased and there are various much better measures, e.g. realised volatility or range-based estimators if one does not want to engage with high-frequency data.

Zheng et al (2014) found that the use of absolute values was comparable to realised volatility as a measure of market risk. They used data from the Tokyo Stock Exchange on the 30 stocks of the TOPIX Core30 Index and found that both enable strong predictions of future market behaviour and are similarly sensitive metrics. The absolute measure was preferred as it obviates the need to subjectively decide on an adequate sampling intra-day frequency. An inappropriate choice of sampling frequency in the context of the bitcoin market would have led the realised volatility measure to be biased by microstructural noise.

Zheng et al (2014) is now referenced in the revised paper (see Section 2.1).

- Under Eq. 3.1, it is not completely clear whether the ADF tests were run on daily percentage changes or the original data.

The ADF tests were run on daily percentage changes (see end of paragraph 2 under equation 3.1). We have made this point clear through adding to the end of the third paragraph after equation 3.1: "These tests were run on the daily percentage change in value with the exception of the 'Binary Price Variable', which was already based on the change in price."

- Numbers in Table 2 are very hard to read. I suggest using standard decimal numbers up to some reasonable level, say 4, and then only something like " <0.0001 ", mostly for p-values.

This has been implemented in the revised Table 2.

- There is a huge number of sections and mainly subsections (and sub subsections) that are quite disturbing for a reader. Some restructuring might be helpful.

In the new structure, section numbering is capped to two levels (e.g. Section 1 and Section 1.1).

Reviewer: 1

Comments to the Author(s)

Copyright: "2014TheAuthors" 2019?

Abstract: "words that impact on the metric" should be "words that impact the metric"

This has been implemented.

(Suggestion only) Section 1.1. Perhaps updating the dataset would be worthwhile in getting the latest insights into the paper. This is more important in crypto-asset industry given the volatile nature of the said.

This paper examines changes in word frequency across the 2017-18 price cycle for which clear stages of rising/falling/stable prices can be identified. We lack such clear stages for the 2019 price cycle, as it is still developing. At the time of writing, prices are fluctuating around 10,000 US Dollars; prices may go on to rise further, continue a sideways movement or commence a period of decline. The lack of a clear pattern makes it impossible to perform robust analyses on the data.

Unclear why Figure 2 is Bitcoin Cash and not Bitcoin? Same comment for Table 2. Shouldn't it be Bitcoin? BTC and BCH aren't the same. As of now, it creates confusion and doesn't create a legitimate comparison. The conclusions drawn in the paper must be justified by all analyses.

Apologies for the confusion; we have updated the Google Search data preparation section (Section 2.2) for clarity. We are not examining Google searches for 'Bitcoin Cash' but rather for the topic of bitcoin, which Google Trends labelled, at the time of data collection, 'Bitcoin – Cash'. Figure 2 and Table 2 have been amended accordingly.

Replace tables with figures for Table 3 and Table 4. Visualising the transition from stage to stage will be a value add. This is to justify the conclusions drawn.

This has been implemented; please see the new figures 3 and 4.

Consider publishing the data for further research and repeatability.

The dataset and code used have been uploaded to the Dryad depository. Please see: <https://datadryad.org/review?doi=doi:10.5061/dryad.8n6m564>.

This is stated in the amended Data Accessibility Statement and at the beginning of Section 2.